# Blue light and $CO_2$ signals converge to regulate light-induced stomatal opening

Asami Hiyama[1], Atsushi Takemiya[1,5], Shintaro Munemasa [2], Eiji Okuma [2], Naoyuki Sugiyama[3], Yasuomi Tada [4], Yoshiyuki Murata [2] & Ken-ichiro Shimazaki [1]

Stomata regulate gas exchange between plants and atmosphere by integrating opening and closing signals. Stomata open in response to low $CO_2$ concentrations to maximize photosynthesis in the light; however, the mechanisms that coordinate photosynthesis and stomatal conductance have yet to be identified. Here we identify and characterize CBC1/2 (CONVERGENCE OF BLUE LIGHT (BL) AND $CO_2$ 1/2), two kinases that link BL, a major component of photosynthetically active radiation (PAR), and the signals from low concentrations of $CO_2$ in guard cells. CBC1/CBC2 redundantly stimulate stomatal opening by inhibition of S-type anion channels in response to both BL and low concentrations of $CO_2$. CBC1/CBC2 function in the signaling pathways of phototropins and HT1 (HIGH LEAF TEMPERATURE 1). CBC1/CBC2 interact with and are phosphorylated by HT1. We propose that CBCs regulate stomatal aperture by integrating signals from BL and $CO_2$ and act as the convergence site for signals from BL and low $CO_2$.

[1] Department of Biology, Faculty of Science, Kyushu University, 744 Motooka, Fukuoka 819-0395, Japan. [2] Graduate School of Environmental and Life Science, Okayama University, Okayama 700-8530, Japan. [3] Department of Molecular & Cellular BioAnalysis, Graduate School of Pharmaceutical Sciences, Kyoto University, Sakyo-ku, Kyoto 606-8501, Japan. [4] Center for Gene Research, Nagoya University, Chikusa, Nagoya 464-8602, Japan. [5] Present address: Graduate School of Sciences and Technology for Innovation, 1677-1 Yoshida, Yamaguchi 753-8512, Japan. Correspondence and requests for materials should be addressed to K.-i.S. (email: kenrcb@kyushu-u.org)

Stomata in plants open in response to light and low concentrations of $CO_2$ and facilitate $CO_2$ uptake from the atmosphere for photosynthetic $CO_2$ fixation and the transpirational stream that delivers mineral nutrients to plant tissues through xylem[1–4]. Stomata close in response to drought, $Ca^{2+}$, abscisic acid (ABA), other plant hormones, and high concentrations of $CO_2$ to prevent water loss[5–9]. Optimal stomatal apertures are maintained by integrating these ever-changing and antagonizing factors; appropriately sized apertures maximize plant growth[1, 8]. Furthermore, a close correlation between photosynthetic $CO_2$ fixation and stomatal conductance is often found under various light intensities, but the underlying mechanisms for this relationship are unknown[4, 10].

The opening of stomata is induced by blue light (BL), red light (RL), and low concentrations of $CO_2$. Stomatal opening responses specific to BL are enhanced by RL and are ubiquitous in land plants except the fern species belonging to the Polypodiopsida[3, 11]. All of the signaling components for the BL response exist in guard cells, and guard cell protoplasts (GCPs) swell in response to BL[12]. BL is perceived by phototropins (phot1, phot2)[13], plant-specific BL receptor kinases[14, 15]. After autophosphorylation of phototropins[16–18], the signals activate plasma membrane $H^+$-ATPases via the signaling components[19] that include BLUS1 kinase and a type1 protein phosphatase[20, 21]. $H^+$-ATPase activation is induced by phosphorylation of the penultimate threonine in the C terminus[19], and the $H^+$-ATPase hyperpolarizes the membrane potential by pumping $H^+$ out of the guard cells[22–24]. The hyperpolarization drives $K^+$ uptake through the inward-rectifying $K^+$ ($K^+_{in}$) channels[25] with accumulation of malate[2−], $Cl^−$, and $NO_3^{− 3}$. Simultaneous activation of $K^+_{in}$ channels in the phototropin-mediated pathway is also known[26]. Malate is synthesized in guard cells via starch degradation that is induced by the activated $H^+$-ATPase[27]. The accumulated $K^+$ salt in guard cells decreases the water potential and causes water uptake, resulting in stomatal opening. The signals from phototropins also inhibit the S-type anion current[28] that may support stomatal opening; however, the signaling components connecting the phototropins, the $H^+$-ATPase, and the anion channels remain unknown.

In contrast to BL-dependent stomatal opening, the mechanisms by which RL induces stomatal opening are a matter of debate[2–4, 29–31]. RL causes membrane hyperpolarization in guard cells of *Vicia faba*[32] and stomatal opening in the epidermal peels of *Vicia* and *Commelina*. Both of these responses to RL are suppressed by DCMU, a photosynthetic electron transport inhibitor[33]. RL-induced stomatal opening is not evident in epidermal peels of *Arabidopsis* plants[16, 21], but opening is conspicuous in the intact leaves. In line with this observation, recent investigations have suggested that in the light diffusible substances are moved from mesophyll cells to guard cells to induce leaf stomatal opening[30, 34], but the substances have yet to be identified. RL-induced stomatal opening may be caused by a low intercellular concentration of $CO_2$ ($Ci$) brought about by mesophyll photosynthesis because low $CO_2$ has been reported to cause stomatal opening[2, 27, 35]. In contrast, other investigators have reported that such a reduction in the $Ci$ of leaves was insufficient to cause stomatal opening[4, 30].

Closure of stomata is driven by the release of anions from guard cells via the S-type channels of SLAC1 (SLOW ANION CHANNEL ASSOCIATED 1) and SLAH3 (SLAC1 HOMOLOGUE 3)[36–38]. These channels are activated in response to ABA, $Ca^{2+}$, and high concentrations of $CO_2$[5, 9, 39]. The release of anions across the membrane causes depolarization, resulting in activation of outward-rectifying $K^+$ channels and, thereby, induces stomatal closure through the simultaneous efflux of anions and $K^+$[2, 26]. Inhibition of the $H^+$-ATPase also occurs in response to ABA or $Ca^{2+}$ and maintains membrane depolarization[40–42]. With respect to $CO_2$-specific stomatal closure, carbonic anhydrases[43], a MATE-type transporter RHC1 (RESISTANT TO HIGH $CO_2$)[44] and HT1 (HIGH LEAF TEMPERATURE 1)[45] kinase are responsible for transducing the high $CO_2$ signal in this order and activate the S-type anion channels via OST1 (OPEN STOMATA 1), a common signal transducer for ABA and $CO_2$[46]. Recently, two mitogen-activated protein kinases (MPKs), MPK4 and MPK12, are shown to inhibit HT1 activity, thereby, stimulating stomatal closure in response to $CO_2$[47, 48]. Two carbonic anhydrases, βCA1 and βCA4, stimulate stomatal closure by converting $CO_2$ to bicarbonate[43]. In contrast, HT1 kinase functions as a negative regulator of $CO_2$-induced stomatal closure. In *ht1-1* and *ht1-2* mutants, stomatal opening was impaired or disrupted in response to low concentrations of $CO_2$, and both mutants exhibited constitutive high $CO_2$ responses[44, 45, 49]. RHC1 transduces the $HCO_3^−$ signal into suppressing HT1[44] and activates OST1 by bridging the signaling between the CAs and HT1, resulting in anion channel activation[44]. Phosphorylation of the N terminus of SLAC1 by OST1 is required for ABA-induced stomatal closure[39, 50], but a recent report suggested that phosphorylation of the transmembrane domain in SLAC1 is crucial to $CO_2$-induced stomatal closure[51]. Furthermore, HT1 is hypothesized to have an essential role in RL-induced stomatal opening under the low $Ci$ via the interaction with OST1[49, 52], but the functional roles of HT1 in $CO_2$ signaling and RL-induced stomatal opening are unclear.

In this study, we searched for proteins that are phosphorylated in response to BL using phosphoproteome analyses to identify missing components in the phototropin-mediated signaling pathway in guard cells. We found that a novel protein kinase is rapidly phosphorylated in a phototropin-dependent manner and that the kinase and its homolog are redundantly responsible for light-induced stomatal opening. We demonstrated that the protein kinases mediate the inhibition of the S-type anion channels in response to BL and also provide evidence that the kinases are required for stomatal opening in response to low concentrations of $CO_2$ and function in the same signaling pathway as HT1.

## Results

**Phosphoproteome analyses of proteins in guard cells.** To identify the signaling components in the pathway for phototropin-mediated stomatal responses, we used phosphoproteome analyses of guard cell protoplasts (GCPs) from *Arabidopsis thaliana*. The method facilitates the identification of components that function redundantly because the mutant screening with a single mutation does not always provide clear phenotype. GCPs were illuminated with a short pulse of blue light (BL, $100\ \mu mol\ m^{-2}\ s^{-1}$) for 30 s superimposed on a background of high-fluence-rate red light (RL, $600\ \mu mol\ m^{-2}\ s^{-1}$). Strong RL maintains photosynthesis in GCPs at near saturation and enables to isolate the response specific to BL. We stopped the reaction at 0.5 and 2.5 min after the start of BL pulse by adding trichloroacetic acid (TCA) to GCPs and collected GCP samples because phosphorylation levels of phototropin and the $H^+$-ATPase, respectively, reach their maximum at these times[16, 53]. The samples were digested, and the resulting phosphopeptides were subjected to mass spectrometric analyses[54]. We selected phosphopeptides that were rapidly phosphorylated within 30 s after BL exposure in a phototropin-dependent manner (Fig. 1a, b). Phosphopeptides having the sequence "RKpSLpSDGEDNVNNTR" were derived from the gene product of At3g01490 that encodes a novel Ser/Thr protein kinase with a deduced molecular mass of 47 kDa and has a typical kinase domain in the C terminus. The

amount of this protein did not change in response to BL for 30 s (Fig. 1b, inset). The phosphopeptides were reported in PhosPhAt 4.0 database. The At3g01490 gene comprises 411 amino acids with no transmembrane domain (Fig. 1c) and was classified as a mitogen-activated protein kinase kinase kinase (MAPKKK, ref. [55]). The protein was named CBC1 (CONVERGENCE OF BL AND $CO_2$ 1) kinase because CBC1 transduces the signals not only from BL, but also from $CO_2$ into stomatal movement; further justification for this name will be described below. The analyses also revealed that the phototropins and the $H^+$-ATPase isoforms of AHA1, 2, 4 (or 11), and 5 (or 8) were phosphorylated in response to BL (Supplementary Fig. 1), as has

been reported[16–18], and enabled the simultaneous identification of phosphorylation sites in multiple proteins.

**Stomatal opening is impaired in the *cbc1 cbc2* double mutant.** To identify the role of CBC1 kinase in BL-dependent stomatal opening, we obtained a T-DNA insertion mutant (*cbc1*) (Fig. 1c). The temperature of wild-type (WT) plants measured by infrared thermography decreased in response to BL, a result of stomatal opening[21], but the temperature of the *phot1-5 phot2-1* mutant did not decline (Fig. 1d, e). The *cbc1* mutant was slightly impaired in the temperature decrease. Since CBC1 belongs to subgroup C7 of the MAPKKK family (ref. [55] and Supplementary Fig. 2), which

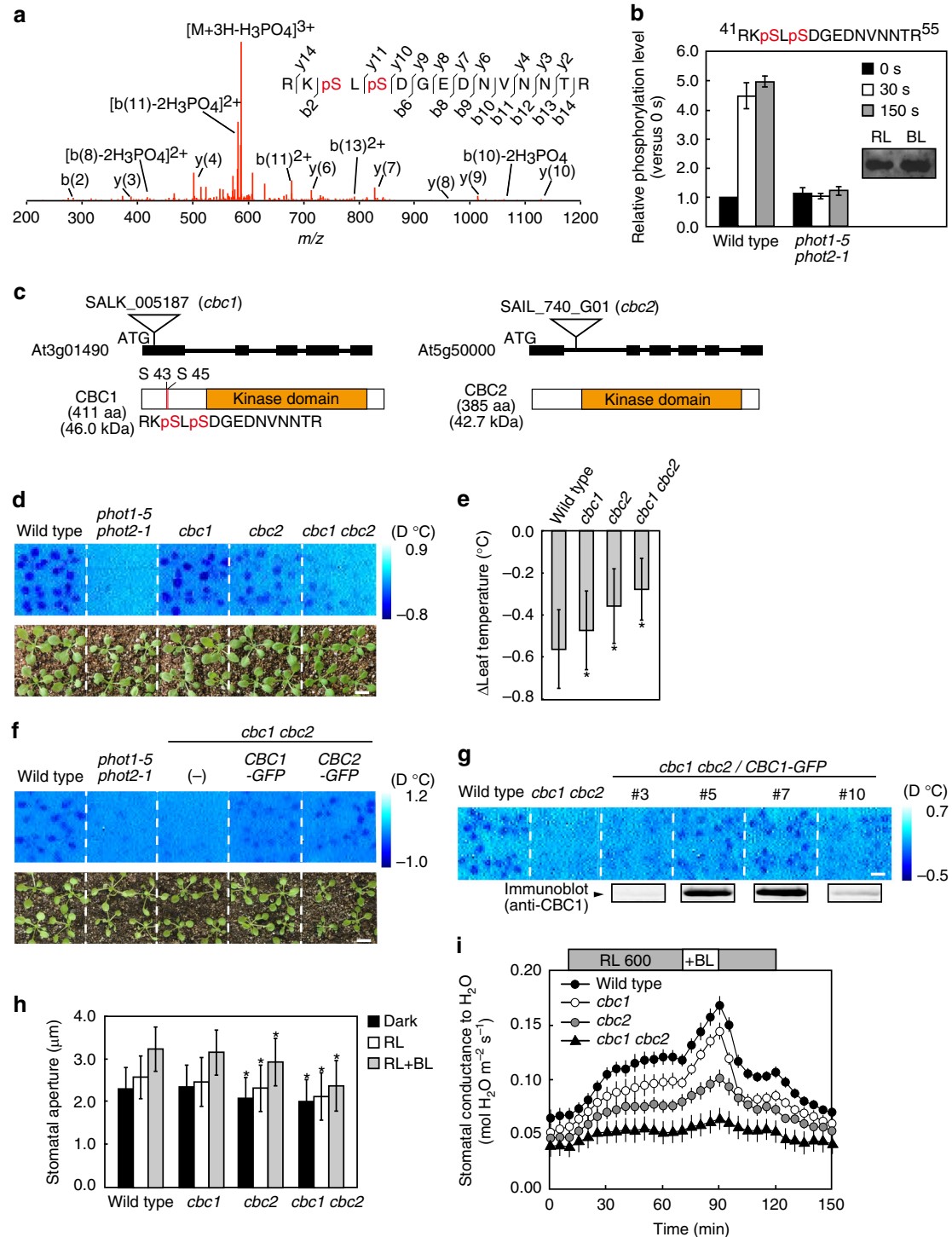

includes four other genes, we selected At5g50000 in the same clade of *CBC1* and named it *CBC2*. *CBC1* and *CBC2* genes were both expressed in guard cells (Supplementary Fig. 3a). When *CBC1-GFP* or *CBC2-GFP* was expressed in the *cbc1 cbc2* double mutant, GFP fluorescence was found in the cytosol of guard cells (Supplementary Fig. 3b). A T-DNA insertion mutant of At5g50000 (*cbc2*) exhibited partial impairment in the temperature decrease (Fig. 1d, e). The double mutant (*cbc1 cbc2*) was severely impaired in the temperature decrease. The impairment was partially complemented by transforming the mutant with *CBC1-GFP* or *CBC2-GFP* (Fig. 1f). When *CBC1-GFP* was overexpressed in *cbc1 cbc2* using each own promoter, the temperature decrease was restored almost completely in proportion to the expression levels of *CBC1-GFP* (Fig. 1g). These results suggest that CBC1 and CBC2 have redundant roles in BL-dependent stomatal opening.

In epidermal peels, stomata scarcely opened in response to RL (50 μmol m$^{-2}$ s$^{-1}$) but did substantially to BL (10 μmol m$^{-2}$ s$^{-1}$) superimposed on RL in WT (Fig. 1h). BL-dependent stomatal opening was not affected in *cbc1* and slightly impaired in *cbc2*, but severely in *cbc1 cbc2*. Stomatal apertures were smaller in *cbc1 cbc2* than in WT under darkness, with apertures of 2.55 ± 0.10 for the mutant and 2.89 ± 0.11 μm for WT ($P = 2.34 \times 10^{-6}$) (Supplementary Table 1a). Since no difference was found in stomatal size between WT and the mutant (Supplementary Table 1b), stomata closed tighter in *cbc1 cbc2* than in WT plants.

In intact leaves, RL at 600 μmol m$^{-2}$ s$^{-1}$ induced an increase in stomatal conductance that reached a steady state within 60 min (Fig. 1i). BL at 20 μmol m$^{-2}$ s$^{-1}$ superimposed on the RL induced a rapid increase in stomatal conductance. The response is specific to BL and is distinct from photosynthesis-dependent stomatal opening[22, 56]. Stomatal opening by both RL and BL was partially reduced in both the *cbc1* and *cbc2* single mutants and was severely reduced in the *cbc1 cbc2* double mutant (Fig. 1i and Supplementary Table 2). To see this easily, the opening responses were normalized to the same starting values on the basis of raw data in Fig. 1i (Supplementary Fig. 4). From these results, we conclude that CBC1 and CBC2 redundantly function to open stomata in response to light. We note that stomata in leaves closed tighter in the double mutant than in WT (Fig. 1i).

We note that stomata exhibited the different responses between leaves and epidermis in *Arabidopsis*, with opening in leaves and negligible opening in epidermis under RL (Fig. 1h, i). Such difference is partly due to the situation where guard cells are placed. In leaves, mesophyll tissues might provide guard cells with unidentified substances that stimulate stomatal opening but in epidermal peels such tissues are absent.

We investigated BL signaling pathway that mediates stomatal opening in the double mutant. The pathway includes the signaling components of phot1, phot2, BLUS1 kinase, and H$^+$-ATPase. Unexpectedly, neither of the components nor light signaling was affected in *cbc1 cbc2* (Supplementary Fig. 5). Autophosphorylation of phot1 and phot2[16, 17], phosphorylation of BLUS1[21] and H$^+$-ATPase[19] were not impaired (Supplementary Fig. 5a–c). Proton pumping in GCPs in response to BL and fusicoccin (Fc), an activator of the H$^+$-ATPase, occurred normally in *cbc1 cbc2* (Supplementary Fig. 5d). We thus hypothesized that the mutation impairs a function downstream of the H$^+$-ATPase, and K$^+_{in}$ channels are candidates. Whole-cell K$^+_{in}$ current in GCPs determined by patch-clamp technique was reduced by 50% in *cbc1 cbc2* (Supplementary Fig. 5e). However, such a reduction in K$^+_{in}$ current will not inhibit stomatal opening because the K$^+_{in}$ current in guard cells is sufficiently large for opening to occur[57, 58]. In accord with this interpretation, stomata in the double mutant opened with a similar time course to those of WT in response to Fc (Supplementary Fig. 5f), but with a slight reduction of opening.

**S-type anion channels cause reduced stomatal opening**. The mechanisms generating a driving force for BL-dependent stomatal opening so far identified were not impaired in the *cbc1 cbc2* double mutant, but light-induced stomatal opening was reduced. Such reduction is probably due to accelerated closing processes. During the measurement of stomatal apertures of light-treated epidermal peels by microscope, we noticed that opened stomata gradually closed in the mutant, but such a response was not found in WT. When 50 mM KCl was present in the bathing buffer (pH 6.5), once opened stomata remained constant in the *cbc1 cbc2* double mutant (Fig. 2a), but gradually closed in *cbc1 cbc2* when KCl was removed. We thus suspected that the S-type anion channels might release Cl$^-$ from mutant guard cells even under light. To test this hypothesis, we investigated the effect of an anion channel blocker, anthracene-9-carboxylic acid (9-AC, ref. [59]), on stomatal movement and found that stomatal closure in the mutant was arrested by 9-AC without KCl (Fig. 2b). The result suggests that S-type channels release Cl$^-$ in *cbc1 cbc2* under light and causes reduced stomatal opening.

To show genetically the involvement of S-type anion channels of SLAC1 and/or SLAH3[36–38] in the reduced stomatal opening, we generated the triple mutants of *cbc1 cbc2 slac1-4* and *cbc1 cbc2 slah3-3*. Stomatal apertures were larger in *slac1-4* than in WT under both darkness and light (Fig. 2c). The apertures were

**Fig. 1** Identification of phosphorylated proteins in guard cells mediated by phototropins using phosphoproteome analyses. **a** MS/MS spectrum of the phosphopeptide having the sequence RKpSLpSDGEDNVNNTR. **b** Changes in the phosphorylation levels of proteins having RKpSLpSDGEDNVNNTR in response to BL. Amounts of the phosphopeptides were calculated from the integrated peak values. The samples were obtained just before the BL, 0.5 and 2.5 min after the start of BL. We used four replicates of GCP preparations. Bars represent means ± s.e.m. Inset: Determination of the amount of this protein by immunoblot just before BL and 0.5 min after the pulse of BL. **c** Upper: Genomic structures of *CBC1* and *CBC2* genes and mutation sites of *cbc1* and *cbc2* shown by triangles as T-DNA insertion sites. Black boxes and lines indicate exons and introns, respectively. Lower: Schematic structures of CBC1 and CBC2. Red lines indicate the phosphorylated residues in CBC1. Orange boxes indicate the kinase domains. **d, f, g** Thermal images of *Arabidopsis* leaves from WT and *phot1-5 phot2-1*, *cbc1*, *cbc2*, *cbc1 cbc2* (**d**); from WT and the *phot1-5 phot2-1*, *cbc1 cbc2* mutants, and transgenic plants of *cbc1 cbc2* expressing *CBC1-GFP* and *CBC2-GFP* (**f**); from WT and transgenic plants of *cbc1 cbc2* expressing *CBC1-GFP* at different levels (**g**); in response to BL. Images were obtained by subtracting the image taken before the BL pulse from the image taken 15 min after the pulse. Plants were illuminated with RL (80 μmol m$^{-2}$ s$^{-1}$) for 50 min and then a weak pulse of BL (5 μmol m$^{-2}$ s$^{-1}$) was superimposed for 15 min. The blue gradation bar displays the temperature decrease by BL. The white bar represents 1 cm. **e** Quantitative data on the temperature decreases from thermal images in WT, *cbc1*, *cbc2*, and *cbc1 cbc2*. Bars represent means ± s.e.m. (*n* = 12, pooled from triplicate experiments). **h** Light-dependent stomatal opening in the epidermes from WT and *cbc* mutants. Epidermes were illuminated with RL (50 μmol m$^{-2}$ s$^{-1}$) and BL (10 μmol m$^{-2}$ s$^{-1}$) for 2 h. Stomatal apertures were measured for 25 stomata from epidermal peels floating on water. Experiments were repeated three times in different occasions. Bars represent means ± s.d. (*n* = 75). Asterisks denote the significant differences in stomatal apertures between WT plants and mutants under darkness, RL, and RL + BL. *$P$ < 0.05 by Student's *t* test. **i** Light-induced stomatal opening determined by stomatal conductance. Plants were illuminated with RL (600 μmol m$^{-2}$ s$^{-1}$) for 80 min, then BL (20 μmol m$^{-2}$ s$^{-1}$) was superimposed for 20 min. Bars represent ± s.e.m. (*n* = 8)

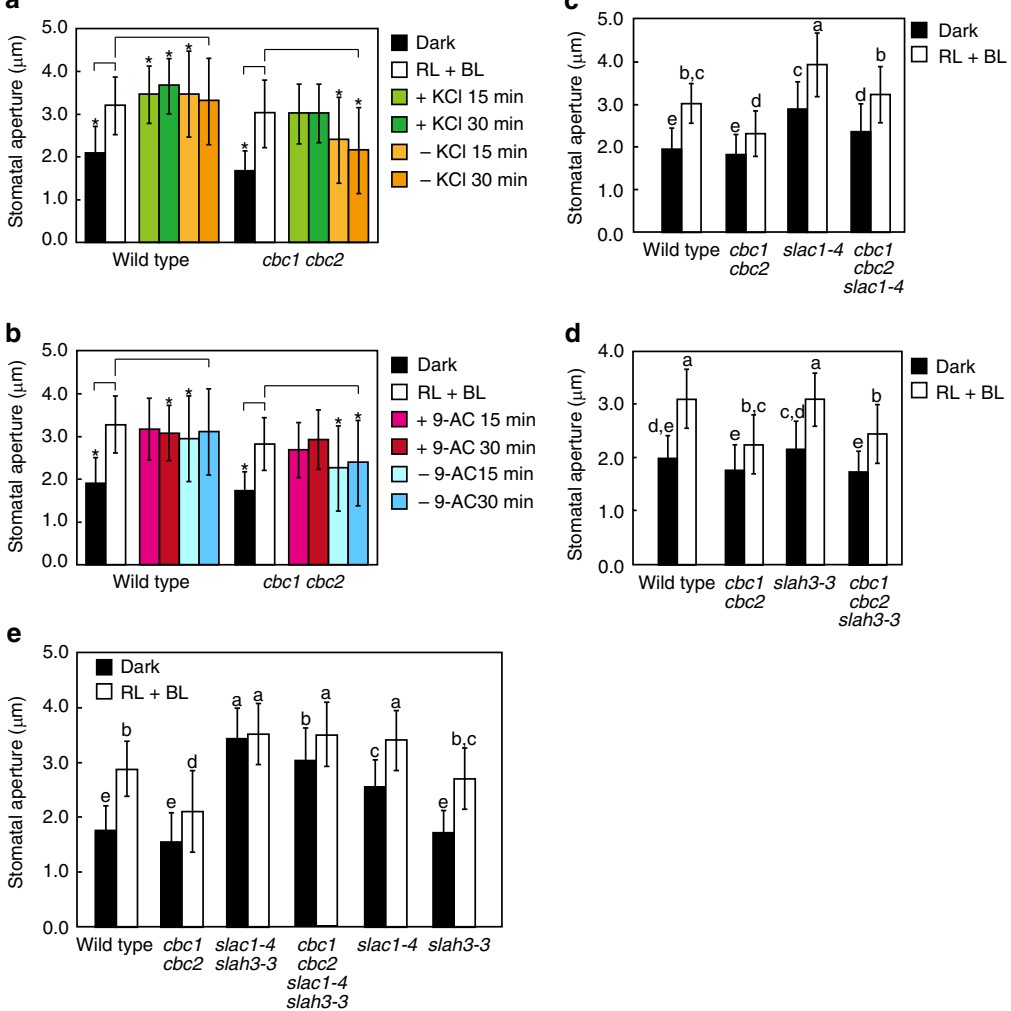

**Fig. 2** Enhancement of stomatal closure in the *cbc1 cbc2* mutants in the absence of KCl. **a** Light-induced stomatal opening in WT and *cbc1 cbc2*. Epidermal peels in 5 mM MES-bistrispropane buffer (pH 6.5) containing 50 mM KCl were illuminated with RL ($50 \, \mu\text{mol} \, \text{m}^{-2} \, \text{s}^{-1}$) plus BL ($10 \, \mu\text{mol} \, \text{m}^{-2} \, \text{s}^{-1}$) for 2 h to open stomata. Stomatal apertures were measured from epidermal peels floating on the buffer. A portion of the peels was transferred to a buffer containing 50 mM KCl or no KCl and held for 15 or 30 min in the light; epidermal peels were left floating on the same buffer for measurement of stomatal apertures. Asterisks denote the significant differences in stomatal apertures in comparison with WT or *cbc1 cbc2* double mutants treated with RL + BL. $*P < 0.05$ by Student's *t* test. **b** Inhibition of stomatal closure by anthracene-9-carboxylic acid (9-AC) in the absence of KCl. Once stomata were allowed to open as described in **a**, then epidermal peels were transferred to a KCl-free buffer containing 100 μM 9-AC. **c**, **d**, **e** Light-induced stomatal opening in WT, *cbc1 cbc2*, *slac1*, and *slah3*, their triple (*cbc1 cbc2 slac1-4* and *cbc1 cbc2 slah3-3*) and quadruple (*cbc1 cbc2 slac1-4 slah3-3*) mutants. Significant differences are denoted with different lowercase letters and analyzed by ANOVA tests followed by a post hoc Tukey test. Stomatal apertures were measured for 25 stomata from floating epidermal peels, and the experiments were repeated three times in different occasions. Bars represent means ± s.d. (*n* = 75, pooled from triplicate experiments)

smaller in *cbc1 cbc2 slac1-4* than in *slac1-4*, probably due to the activation of SLAH3 in *cbc1 cbc2 slac1-4*. Likewise, stomatal apertures were smaller in *cbc1 cbc2 slah3-3* than in *slah3-3* (Fig. 2d), caused by activated SLAC1 in *cbc1 cbc2 slah3-3*. We generated the quadruple mutant *cbc1 cbc2 slac1-4 slah3-3*. Stomatal apertures were much larger in *slac1-4 slah3-3* than in WT (Fig. 2e). The apertures in the quadruple mutant were much larger than those in *cbc1 cbc2* and were almost similar to those in *slac1-4 slah3-3*. The results indicate that the *SLAC1* and *SLAH3* genes are epistatic to the *CBC1* and *CBC2* genes, suggesting that SLAC1 and SLAH3 are regulated by CBC1 and CBC2. Loss of CBC function might fail to inhibit S-type anion channels under the light. Furthermore, the stomatal apertures in *slac1-4* and *slah3-3* were smaller than those in *slac1-4 slah3-3* (Fig. 2c–e), suggesting the redundant function of SLAC1 and SLAH3 in stomatal closure. In most cases, however, the stomatal apertures

in *slac1-4* were larger than those in *slah3-3*, suggesting that the SLAH3 role in stomatal closure under our conditions was not large in our materials. We note that light-enhanced stomatal opening in these plant materials, at least be partly caused by the pump activation (Fig. 2c–e and Supplementary Fig. 5).

**CBCs regulate the S-type anion channels in response to BL.** BL inhibits the S-type anion currents in guard cells via phototropins in *Arabidopsis* leaves[28]. Our results suggest that gene disruption of both *CBC1* and *CBC2* impairs the inhibitory action of BL, thereby causing a reduction in stomatal opening. To test this hypothesis directly, the S-type anion currents were measured in *Arabidopsis* GCPs by whole-cell patch-clamp technique (Fig. 3). Since a large current was not observed under the conditions, we pre-exposed WT GCPs to extracellular bicarbonate as has been reported[43] and

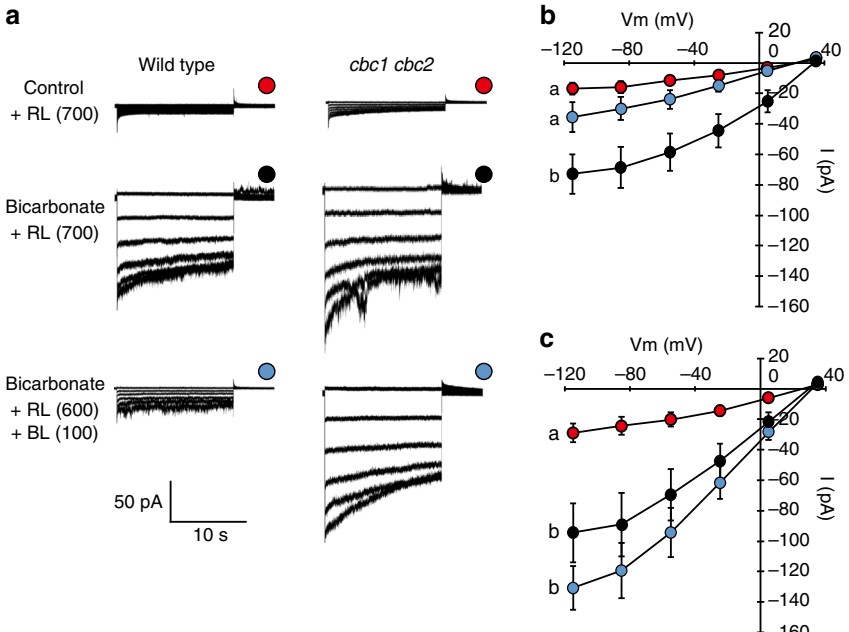

**Fig. 3** The inhibition of bicarbonate-activated S-type anion currents by BL is abolished in *Arabidopsis* guard cells from the *cbc1 cbc2* mutant. **a** Representative whole-cell currents of WT and *cbc1 cbc2*. **b, c** Whole-cell outward S-type anion current–voltage relationships in GCPs of WT (**b**), and *cbc1 cbc2* (**c**). The data are presented as means ± s.e.m. (WT control RL, $n = 7$; WT bicarbonate RL, $n = 7$; WT bicarbonate RL + BL, $n = 6$; *cbc1 cbc2* control RL, $n = 8$; *cbc1 cbc2* bicarbonate RL, $n = 7$; *cbc1 cbc2* bicarbonate RL + BL, $n = 5$). Significant differences are denoted with different lowercase letters. GCPs were irradiated with RL (600 μmol m$^{-2}$ s$^{-1}$) and treated with or without 1.2 mM bicarbonate buffered to 1 mM free $CO_2$ (pH 5.6). To measure BL-dependent S-type anion current inhibition, GCPs were irradiated with RL (600 μmol m$^{-2}$ s$^{-1}$) and BL (100 μmol m$^{-2}$ s$^{-1}$)

confirmed that the S-type anion current was enhanced fourfold (Fig. 3a, b). We found that the current was greatly reduced by illuminating GCPs with BL at 100 μmol m$^{-2}$ s$^{-1}$ under RL (Fig. 3a, b), suggesting that the reduction is BL dependent. The current enhancement by bicarbonate was also documented in *cbc1 cbc2* GCPs (Fig. 3a, c). However, the enhanced current in *cbc1 cbc2* was not reduced by BL. These results suggest that the signals from phototropins negate the $CO_2$-enhanced anion current of the S-type channels, and CBC1 and/or CBC2 mediate the phototropin-dependent response. We also showed that the anion currents in GCPs from WT and *cbc1 cbc2* were similarly enhanced by extracellular 40 mM $Ca^{2+}$ (Supplementary Fig. 6)[39], suggesting that the mutation of CBCs did not affect the $Ca^{2+}$ sensitivity in the S-type channels.

**CBC1 is phosphorylated by phot1.** Phosphoproteome analyses revealed that CBC1 was phosphorylated in the phototropin-mediated pathway (Fig. 1). We indirectly showed the phosphorylation by mobility shift using Phos-tag PAGE and immunological method (Fig. 4a). The CBC1 in GCPs of WT and *cbc2* was shifted upward in response to BL, and the upward shift was phototropin-dependent (Fig. 4b). No CBC1 protein was found in *cbc1* and *cbc1 cbc2*. Antibodies against CBC2 recognized four protein bands, but the CBC2 mobility was not changed (Fig. 4a and Supplementary Fig. 7).

Since CBC1 was rapidly phosphorylated in response to BL (Fig. 1b), CBC1 possibly interacts with phot1 and is directly phosphorylated by phot1. His-CBC1 or His-CBC2 was mixed with FLAG-phot1. Pull-down assays indicated that both CBC1 and CBC2 co-precipitated with phot1, suggesting that both CBCs interacted with phot1 (Fig. 4c). To investigate the interaction in vivo, we conducted bimolecular fluorescence complementation (BiFC) assays in *Arabidopsis* mesophyll protoplasts (MCPs) using CBCs-nYFP and phot1-cYFP (Fig. 4d). Since CBC2-nYFP

expressed in MCPs damaged to the protoplast probably due to kinase activity, we used a kinase-dead CBC2 (D245N) instead of an active CBC2. Co-infection of CBC1 or CBC2 (D245N) with phot1 produced yellow–green fluorescence on the periphery of MCPs, indicating that both CBC1 and CBC2 interacted with phot1. No signal was found in the absence of CBCs, and in the presence of both phot1-cYFP and PIN1-nYFP as a negative interactor for phot1[60] (Supplementary Fig. 8). We next measured the phosphorylation of CBC1 and CBC2 by phot1 in vitro. A glutathione S-transferase (GST)-tagged C-terminal fragment of phot1 (620–996) (P1C) lacking the LOV (light, oxygen, voltage) domains, which is active without BL[61], was combined with kinase-dead CBC1 (D271N) or CBC2 (D245N) as substrates. P1C phosphorylated CBC1 but not CBC2 (Fig. 4e), and the kinase-dead P1C (D806N) did not phosphorylate either substrate.

Phot1 phosphorylated CBC1 in in vitro and in vivo, but it is unclear whether the phosphorylation sites in vitro are the same as those in vivo. To test this, we replaced two in vivo phosphorylated Sers (S43, S45) (Fig. 1c) by Alas in the kinase-dead CBC1 (D271N) and utilized this protein as substrate for P1C. We found no phosphorylation of the CBC1 (S43A, S45A) by P1C (Fig. 4f), suggesting that the phosphorylation sites in vitro are the same as those in vivo or contain one of them.

We interpreted already that the CBC1 mobility shift was caused by phosphorylation in vivo through Phos-tag PAGE (Fig. 4a, b). If so, we can expect that the CBC1 (S43A S45A) will not show such shift in response to BL in vivo. In accord with this, *CBC1* (S43A S45A)-*GFP* expressed in the *cbc1 cbc2* double mutant did not show the shift although WT CBC1-GFP exhibited the typical shift (Fig. 4g). The result confirms that CBC1 is phosphorylated on Ser43 and Ser45 in guard cells and suggests that the mobility shift is caused by phosphorylation of these sites (Fig. 4a, b). We then investigated the interactions between CBCs. Pull-down assays indicated that both CBC1 and CBC2 interacted with either CBC1 or CBC2 (Fig. 4h).

Phot1 directly phosphorylated CBC1 in vitro. Since phototropins directly phosphorylate BLUS1 as in vivo substrate[21], we investigated the role of BLUS1 in the CBC1 phosphorylation. We found that CBC1 was phosphorylated in the *blus1-3* mutant in response to BL (Fig. 4i), suggesting that the BLUS1 does not mediate the signaling between phototropins and CBC1.

**CBC1 and CBC2 function in the same pathway as HT1.** CBCs abolish the $CO_2$-induced activation of S-type anion channels via a phototropin-mediated pathway (Fig. 3). Since stomatal closure by high $CO_2$ concentrations requires the S-type anion channel activation, CBCs function as negative regulators for $CO_2$-induced stomatal closure (Fig. 3c). As HT1 kinase functions as a negative regulator for $CO_2$-induced stomatal closure[44–49], CBCs may act in the same pathway as HT1. We thus investigated stomatal responses to $CO_2$ in the *cbc1 cbc2* double mutant in comparison with those in the *ht1* mutant under darkness. Stomatal conductance increased with the $CO_2$ concentration decrease from

350 to 100 ppm and decreased with the concentration rise 100 to 800 ppm in WT (Fig. 5a). Interestingly, no such response was found in *cbc1 cbc2*. In the *ht1-9* mutant that was identified in our laboratory as having lost kinase activity (Supplementary Fig. 9), stomata had the same responses as those in *cbc1 cbc2* (Fig. 5a). The triple mutant *cbc1 cbc2 ht1-9* revealed the same stomatal phenotype as the *cbc1 cbc2* and *ht1-9* mutants (Fig. 5a). These results indicate that CBC1 and CBC2 function in the same pathway as HT1. Since *ht1* mutants exhibited the normal ABA sensitivity[45], we investigated the ABA sensitivity in the *cbc1 cbc2* double mutant. Stomata closed by ABA in the mutant (Fig. 5b), indicating that CBCs do not play a role in ABA-induced stomatal closure.

We determined the stomatal $CO_2$ sensitivity in the *cbc1* and *cbc2* single mutants under darkness (Fig. 5c). Stomatal conductance increased in both single mutants when the $CO_2$ concentrations were decreased from 350 ppm to 100 or 50 ppm, but the magnitudes were smaller by 50% in the mutants than in

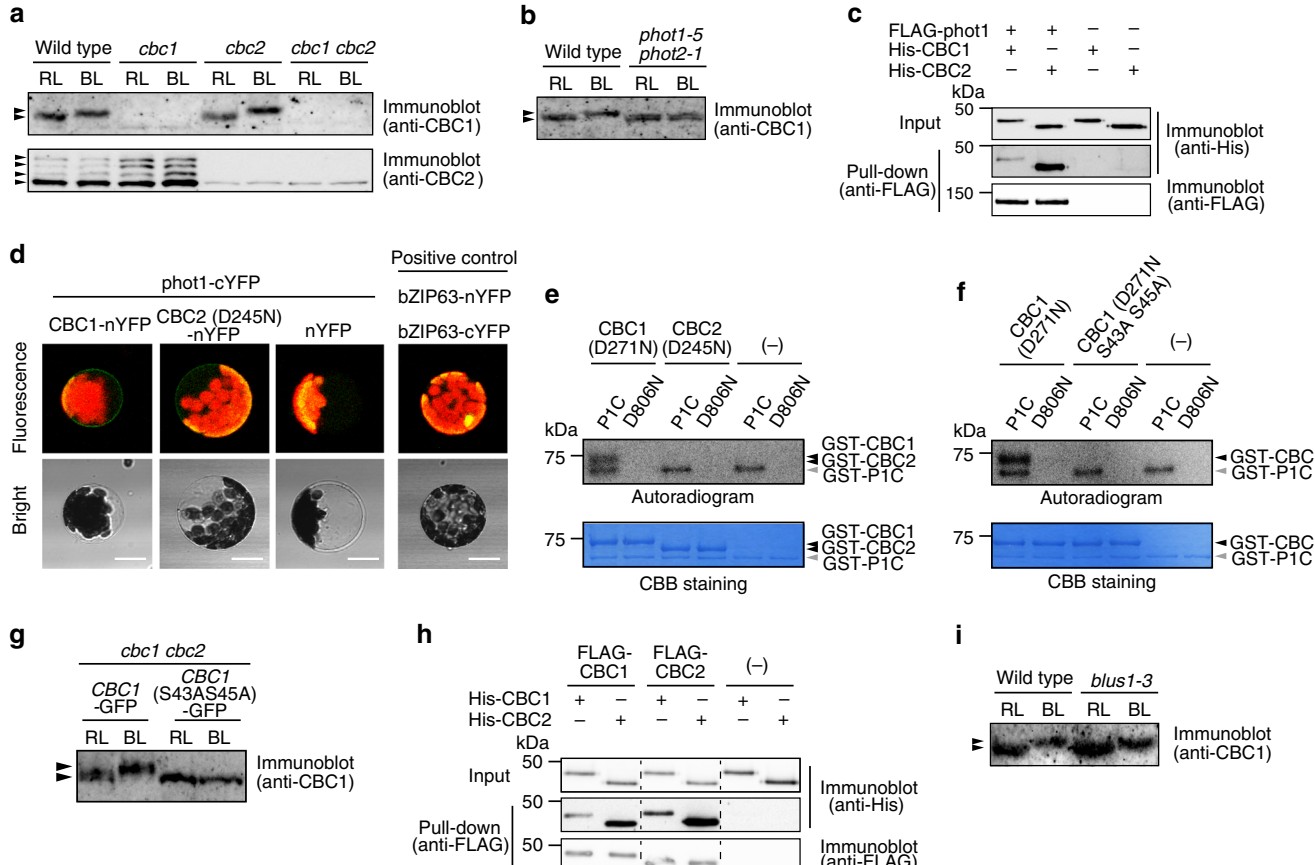

**Fig. 4** Phosphorylation of CBC1 by phototropin1 in a BL-dependent manner. **a** BL-dependent mobility shift of CBC1. GCPs were illuminated with light as described in Fig. 1a. Samples were obtained 1 min after the start of the BL pulse. Immunoblot analysis of CBC1 was performed after electrophoresis on Phostag SDS-PAGE that contained 10% acrylamide, 20 μM Phos-tag, and 40 μM $MnCl_2$. For CBC2, 40 μM Phos-tag and 80 μM $MnCl_2$ were included in the gel. **b** Phototropin-mediated phosphorylation of CBC1. CBC1 detection was done as described in **a**. **c** Pull-down assays of CBC1 and CBC2 by phot1. FLAG-phot1 and His-CBCs were synthesized in vitro transcription/translation. **d** BiFC assays of CBCs and phot1. CBCs-nYFP and phot1-cYFP were co-transfected to MCPs. White bars represent 20 μm. **e** Phosphorylation of kinase-dead CBC1 (D271N) and CBC2 (D245N) by P1C using [γ-$^{32}$P] ATP. Phosphorylation assays were performed in 30 μl reaction mixtures that contained 50 mM Tris-HCl (pH 7.5), 10 mM $MgCl_2$, 3.3 μM ATP, 20 μCi of [γ-$^{32}$P] ATP, and purified proteins. The reactions proceeded for 2 h at 15 °C. GST-tagged P1C or kinase-dead P1C (D806N) were included at 2.5 μg, and CBC1 (D271N) and CBC2 (D245N) were added at 1.2 μg. These GST proteins were expressed in *E. coli* and purified with glutathione-Sepharose beads. **f** Phosphorylation of kinase-dead CBC1 (D271N) and CBC1 (D271N S43A S45A) by P1C. Measurement was done as same as **e**. **g** Mobility of CBC1-GFP and CBC1 (S43A S45A)-GFP expressed in guard cells was determined as **a**, except the acrylamide concentration was used at 6%. The *cbc1 cbc2* double mutant was transformed with *CBC-GFP* or *CBC1 (S43A S45A)-GFP* and GCPs were prepared from these transgenic plants. **h** Pull-down assays of His-CBC1 and His-CBC2 by FLAG-tagged CBC1 and CBC2. Tagged proteins were synthesized by in vitro transcription/translation. **i** BL-dependent phosphorylation of CBC1 in the *blus1-3* mutant. CBC1 detection was done as described in **a**

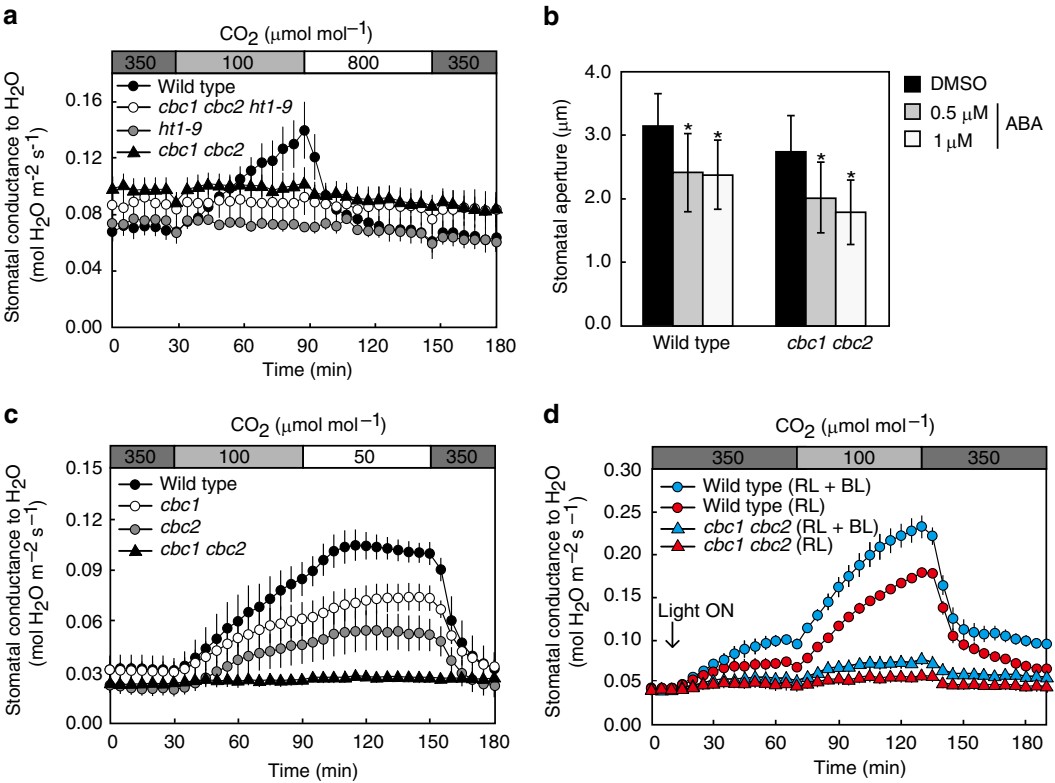

**Fig. 5** Stomatal responses of the *cbc* mutants to $CO_2$ and ABA in *Arabidopsis*. **a** Stomatal movements of the *cbc1 cbc2*, *ht1-9*, and *cbc1 cbc2 ht1-9* mutants in response to changes in $CO_2$ concentration in the dark. **b** Stomatal closure mediated by ABA in *cbc1 cbc2*. **c** Stomatal responses of the *cbc1*, *cbc2* single mutants and the *cbc1 cbc2* double mutant in response to changes in $CO_2$ concentration in the dark. **d** Stomatal conductance increases in WT plants and the *cbc1 cbc2* double mutant in response to the $CO_2$ concentration decrease under the moderate light of RL (90 μmol m$^{-2}$ s$^{-1}$) plus BL (10 μmol m$^{-2}$ s$^{-1}$) or RL (100 μmol m$^{-2}$ s$^{-1}$) only. The $CO_2$ concentration was changed as indicated. **a**, **c**, **d** Bars represent ± s.e.m. ($n = 4$ for **a**, **c** and $n = 8$ for **d**). Stomatal opening in intact leaves was induced by light, and the epidermal peels were obtained. **b** ABA was added to the peels at the indicated concentrations and kept in the light for 2 h. Experiments were repeated three times in different occasions. Bars represent means ± s.d. ($n = 75$)

WT. The results indicate that CBC1 and CBC2 are required for stomatal opening in response to low concentrations of $CO_2$ and function redundantly. We next determined stomatal opening in response to $CO_2$ concentration changes under the moderate light. Stomata opened in WT and slightly in *cbc1 cbc2* in response to RL (90 μmol m$^{-2}$ s$^{-1}$) plus BL (10 μmol m$^{-2}$ s$^{-1}$) (Fig. 5d). Less opening was shown in both plant lines in response to RL (100 μmol m$^{-2}$ s$^{-1}$) than RL plus BL. When $CO_2$ concentration was reduced from 350 to 100 ppm under the light, stomata opened remarkably in WT and only slightly in the double mutant (Fig. 5d). The degree of opening in WT was twofold larger under the light than darkness (Fig. 5c, d). Furthermore, the low $CO_2$-induced opening was larger by 25% under RL plus BL than RL (Fig. 5d and Supplementary Table 3). The results suggest that CBCs substantiate the co-operation between the light and low $CO_2$ for stomatal opening, which enhances the uptake of $CO_2$ for photosynthesis. It is interesting to note here that BL-dependent stomatal opening was enhanced by reduction of the intracellular $CO_2$ concentration in *Commelina*[62]. Taken together, these results indicate that CBC1 and CBC2 integrate and transduce the signals from both BL and $CO_2$ into stomatal movement.

**CBC1 and CBC2 are phosphorylated by HT1 in vitro**. Since CBC1, CBC2, and HT1 function in the same pathway, we expected protein–protein interactions between CBCs and HT1. To investigate this possibility in vivo, we conducted BiFC assays in *Arabidopsis* MCPs using CBCs-nYFP and HT1-cYFP (Fig. 6a). Co-infection of CBC1 or CBC2 (D245N) with HT1 produced

yellow–green fluorescence on the periphery of MCPs, indicating that both CBC1 and CBC2 interacted with HT1. No signal was found without CBCs. We further investigated the interaction between CBCs and HT1 in vitro. FLAG-CBC1 or FLAG-CBC2 was mixed with His-HT1. Pull-down assays indicated that CBC1 and CBC2 co-precipitated with HT1, suggesting that CBC1 and CBC2 physically interacted with HT1 (Fig. 6b). We next determined the protein kinase activities of the CBCs. GST fusions of CBC1 (GST-CBC1) and CBC2 (GST-CBC2) were autophosphorylated (Fig. 6c). A mutation introduced in the kinase domain caused the loss of their autophosphorylation activities in the mutated CBCs (CBC1 (D271N), CBC2 (D245N)). The native CBCs phosphorylated myelin basic protein (MBP) and histone but not casein (Fig. 6d), indicating that CBCs possess protein kinase activity. We then determined whether HT1 could phosphorylate CBC1 and CBC2 and vice versa. Kinase-dead forms of all CBCs or HT1 were used as substrates. We found that HT1 phosphorylated both CBC1 and CBC2, but CBC1 and CBC2 did not phosphorylate HT1 (Fig. 6e). CBC1 and CBC2 may be the substrates of HT1 and likely function downstream of HT1. Since CBC1 can be a substrate for both HT1 and phot1, we tested whether the phosphorylation sites of CBC1 by HT1 are the same as those by phot1. We used CBC1 (D271N S43A S45A) (Figs. 1c and 4f), in which phosphorylatable Sers by phot1 in vivo were substituted by Ala, as a substrate for HT1, and found the phosphorylation of the CBC1 (Fig. 6f). The result suggests that phosphorylation sites in CBC1 by HT1 might be different from those by phot1 or have additional sites, but further investigation will be required.

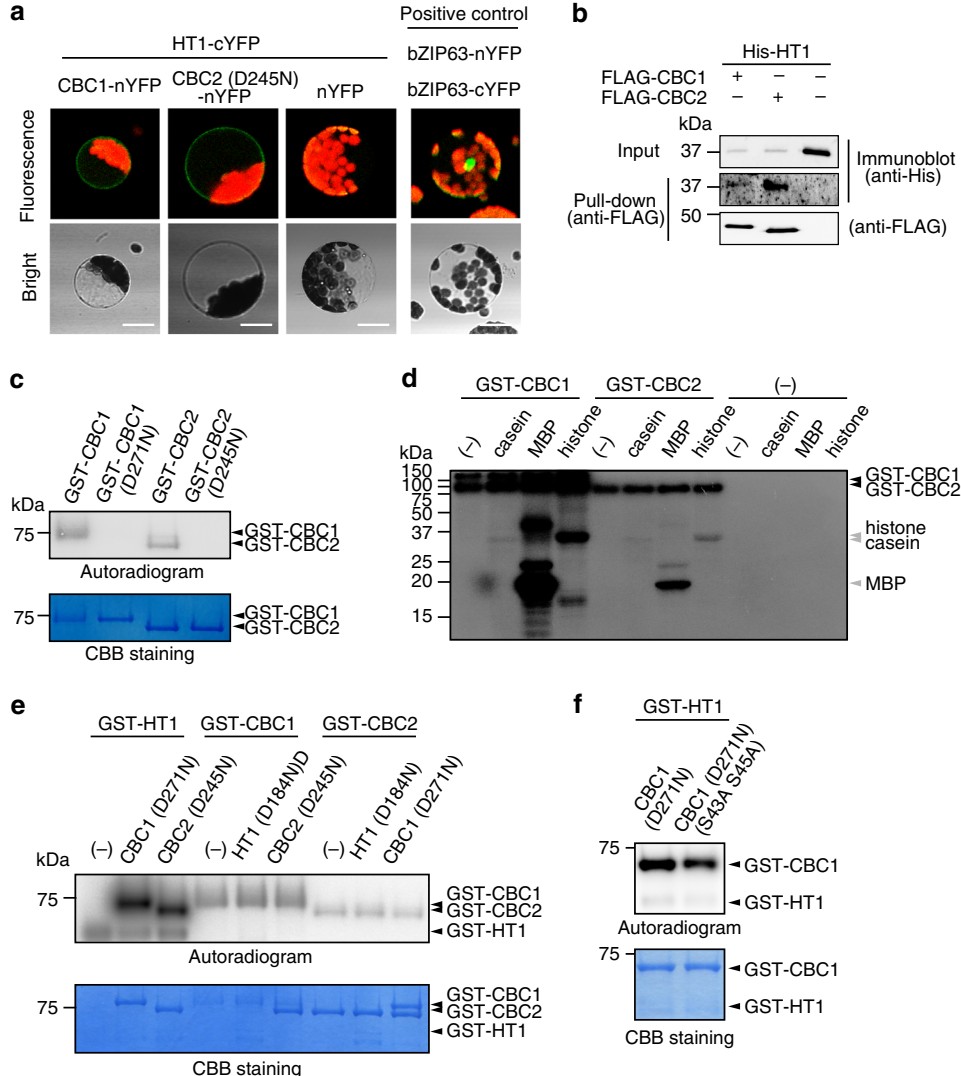

**Fig. 6** CBC1 and CBC2 are phosphorylated by HT1 kinase. **a** BiFC assays of CBCs and HT1. CBCs-nYFP and HT1-cYFP were co-transfected to MCPs. White bars represent 20 μm. **b** Pull-down assays of HT1 by CBC1 and CBC2. FLAG-CBCs and His-HT1 proteins were synthesized in vitro transcription/translation. **c** Autophosphorylation of GST-CBC1 and GST-CBC2. CBC1 (D271N) and CBC2 (D245N) contain the kinase-dead mutation. **d** In vitro kinase assay of CBC1 and CBC2. Artificial substrates of casein, MBP, and histone of 5 μg were used. **e** In vitro kinase assay of HT1, CBC1, and CBC2. Kinase-dead proteins of HT1 (D184N), CBC1 (D271N), and CBC2 (D245N) were used as substrates. One microgram of active kinase and substrate was used. **f** In vitro phosphorylation of CBC1 (D271N) and Ser-substituted CBC1 (D271N S43A S45A) by HT1. All GST-tagged proteins used in **c**–**f** were expressed in *E. coli* and purified with glutathione-Sepharose beads. Phosphorylation assays were done as in Fig. 4e

## Discussion

In this study, we used phosphoproteomic methods to identify phototropin-mediated phosphorylated proteins in guard cells (Fig. 1). Among these proteins, we selected one protein kinase that was phosphorylated in response to BL for detailed study. We named it CBC1 and characterized the role of this protein in stomatal responses. Since a *cbc1* mutant exhibited only slight impairment in the stomatal BL response (Fig. 1d, e), we identified another homologous protein (Supplementary Fig. 2) and named it CBC2. The *cbc2* mutant had slightly impaired stomatal opening by BL and the *cbc1 cbc2* double mutant showed severely impaired response, suggesting that CBC1 and CBC2 function redundantly in stomatal opening (Fig. 1f, g).

How is the stomatal response to BL impaired in the *cbc1 cbc2* mutant? We initially expected that activation of H⁺-ATPase by phototropins was compromised in the mutant. Unexpectedly, neither the components nor the signaling that leads to H⁺-ATPase activation was injured in the mutant (Supplementary

Fig. 5c, d). Since $K^+_{in}$ channel activity was reduced to 50% in *cbc1 cbc2* (Supplementary Fig. 5e), it might cause decreased stomatal opening (Fig. 1). However, the channel activity in guard cells is thought to be much higher than required for the opening in *Arabidopsis*[58, 63, 64]. If the $K^+$ channel activity limited stomatal opening, the opening rate should be decreased in response to fusicoccin as reported[57], but the rate was not affected in *cbc1 cbc2* (Supplementary Fig. 5f). The different results reported are probably caused by different species used[65].

We then suggested that altered activities of S-type anion channels were responsible for stomatal phenotype in the epidermis of *cbc* mutants using various channel mutants (Fig. 2). Furthermore, our patch-clamp experiments revealed that bicarbonate-activated S-type anion channels in GCPs were suppressed in response to BL in WT but not in *cbc1 cbc2* (Fig. 3). From these physiological, genetic, and electrophysiological analyses, we conclude that impaired stomatal opening is due to the inability to inhibit the S-type channels in response to BL and

CBCs mediate the phototropin-dependent inhibition of S-type anion channels. The presence of this pathway was previously reported in intact plants of *Vicia* and *Arabidopsis*[28]; however, our study has now established the physiological relevance of this pathway.

It is evident that CBC1 is located in the phototropin-mediated signaling pathway (Fig. 1). CBC1 physically interacted with phot1 and was directly phosphorylated by phot1 in vitro (Fig. 4), suggesting that CBC1 can be a substrate of phot1. If so, CBC1 should be involved in a pathway distinct from BLUS1 because BLUS1 is a substrate of phototropins[21]. In accord with this, the disruption of *BLUS1* (*blus1-3*)[21] did not affect CBC1 phosphorylation (Fig. 4i). We were unable to show the phosphorylation of CBC2 by phototropins, although CBC2 interacted with phot1 (Fig. 4). Since CBC2 functions redundantly with CBC1 for stomatal opening (Fig. 1), it is unclear how the BL signal is transmitted to CBC2. CBC2 may receive BL signal from CBC1 by protein–protein interaction or from phototropins by the direct interaction because CBC2 interacts with both CBC1 and phot1 (Fig. 4c, h).

We obtained evidence that CBC1 and CBC2 function in the same pathway as HT1 for $CO_2$ signaling (Fig. 5a). For example, the *cbc1 cbc2* double mutant did not respond to $CO_2$ concentration changes. The *cbc1 cbc2* mutant had the same stomatal response as the *ht1-9* single mutant and the *ht1-2* mutant[45] (Fig. 5 and Supplementary Fig. 5f). The *cbc1 cbc2 ht1-9* triple mutant had the same phenotype as *ht1-9*. Furthermore, both CBC1 and CBC2 interacted with HT1 in vivo (Fig. 6a). CBCs physically interacted with HT1 in vitro and were phosphorylated by HT1, but did not phosphorylate HT1 (Fig. 6e). The simple interpretation of these results is that CBCs function downstream of HT1 and ultimately inhibit the S-type anion channels as has been reported[44, 46, 47]. Since CBCs act not only in the phototropin-mediated BL signaling pathway, but also in the HT1-mediated $CO_2$ signaling pathway, we propose a working model indicating that CBCs function where BL and $CO_2$ signals converge (Fig. 7). In this model, red arrows favor stomatal opening in response to BL, RL, and low $CO_2$, and black arrows stimulate stomatal closure in response to high $CO_2$. We noticed here that HT1 (At1g62400) and CBCs belong to the subgroups C5 and C7, respectively, in this phylogenetic tree of the MAPKKK multigene family (Supplementary Fig. 2). These three genes are likely derived from the same ancestral gene. Furthermore, CBC1 and CBC2 recently evolved by duplication.

Recent investigations have indicated that HT1 plays a role in $CO_2$-dependent stomatal closure as a negative regulator[44–49, 52]. We showed that the CBC protein kinases function as negative regulators in the same pathways as HT1 and are involved in inhibiting the activities of SLAC1 and SLAH3. Furthermore, we showed that CBCs were phosphorylated by the HT1 kinase in vitro (Fig. 6). In contrast, HT1 kinase is proposed to directly phosphorylate and inactivate OST1, thereby inhibiting high $CO_2$-induced stomatal closure through inactivation of SLAC1[44]. High concentrations of $CO_2$ abolish HT1 activity and result in the activation of SLAC1 via phosphorylation of the N terminus by active OST1[44, 46] (see Fig. 7). Although CBCs are involved in the $CO_2$ signaling pathways of guard cells, the role(s) and substrate(s) of CBCs are unknown. CBCs may be substrates to inhibit the S-type channels of both SLAC1 and SLAH3 by phosphorylation or indirect regulation via other kinases and phosphatases. Furthermore, high $CO_2$ activates protein kinases other than OST1 and causes $CO_2$-dependent stomatal closure by activation of SLAC1 through phosphorylation of the transmembrane region[51]. CBCs may inhibit the unidentified kinases that activate SLAC1. The reconstitution studies in *Xenopus* oocytes that express these components, including phot1 or HT1, CBCs, S-type channels, and OST1 will confirm these findings in planta.

We showed that RL-induced stomatal opening was impaired to the same degree as BL-dependent stomatal opening in the *cbc* mutants (Fig. 1i and Supplementary Table 2). The lesion in BL-dependent stomatal opening can be accounted by the inability of the mutants to inhibit S-type channels in response to BL (Figs. 2 and 3). In contrast, how RL-induced stomatal opening is inhibited in the mutants is unknown. Since strong RL (also strong BL) reduces $Ci$ by photosynthetic $CO_2$ fixation, we speculated that the reduced $Ci$ might cause stomatal opening by inhibiting the S-type channels through CBCs, but such a response did not occur in the *cbc* mutants. Other mechanisms rather than the inhibition of the S-type channels may exist, and previous reports have suggested that HT1 plays a role in RL-induced stomatal opening[45, 49, 52] and that $H^+$-ATPase is possibly involved in the response[52]. We note that the $H^+$-ATPases can be activated by modifying several sites in their autoinhibitory domain[66]. Further investigation is required for elucidating the role of CBCs in RL-induced stomatal opening.

In conclusion, we showed that CBCs stimulate stomatal opening by inhibiting the S-type anion channels in response to BL, which is a major component of photosynthetically active radiation (PAR). CBCs also likely cause stomatal opening in response to low concentrations of $CO_2$ resulting from photosynthesis by suppressing the S-type channels. We thus propose that CBCs are a convergence point for the signals from PAR and $CO_2$ (Fig. 7). PAR, including both BL and RL, may efficiently support stomatal opening by activating CBCs via phototropins and low $Ci$.

## Methods

**Plant materials and growth conditions**. *Arabidopsis thaliana* plants were grown on soil:vermiculite (1:1) for 4 weeks under a 14/10 h white light (50 μmol m$^{-2}$ s$^{-1}$)/dark cycle at 24 °C. To measure leaf temperature, *Arabidopsis thaliana* seeds were sown on 0.8% (w/v) agar plates containing half-strength Murashige–Skoog salts (pH 5.7), 2.3 mM MES, and 1% (w/v) sucrose and were grown under continuous white light (60 μmol m$^{-2}$ s$^{-1}$) at 23 °C. The 10-day-old plants were transferred to a soil:vermiculite (1:1) mixture and further grown under a 13/11 h light/dark cycle for 7 days. Mutants of *cbc1*, *cbc2*, *blus1-3*[21], *slac1-4*[37], and *slah3-3*[67] were obtained from the Nottingham Arabidopsis Stock Center (NASC), Nottingham, UK. The *phot1-5 phot2-1* mutant has been described in ref. [13].

**Phosphoproteome analyses and phosphopeptide quantification**. Enzymatically isolated GCPs were illuminated with strong RL (600 μmol m$^{-2}$ s$^{-1}$) for 30 min, and then a BL pulse (100 μmol m$^{-2}$ s$^{-1}$) was applied for 30 s. We terminated the reactions at 0.5 and 2.5 min after the start of the pulse by adding TCA to the GCPs. The samples were suspended in 0.1 M Tris-HCl (pH 8.0) containing 8 M urea, protein phosphatase inhibitors, and protease inhibitor cocktails (SIGMA-Aldrich) according to the manufacture's protocol, and sonicated for 5 min. The suspensions were reduced with dithiothreitol, alkylated with iodoacetamide, and digested with Lys-C, followed by tryptic digestion. The digested samples were desalted using StageTips with C18 Empore disk membranes (3 M). Phosphopeptides were enriched by hydroxy acid-modified metal oxide chromatography using lactic acid-modified titania[54]. In brief, the digested samples were diluted with 0.1% TFA, 80% acetonitrile containing 300 mg ml$^{-1}$ lactic acid (solution A), and loaded to custom-made metal oxide chromatography tips preliminary equilibrated with solution A. After successive washing with solution A and 0.1% TFA, 80% acetonitrile, the peptides were eluted with 0.5% piperidine. The obtained fractions were desalted using SDB-XC-StageTips and concentrated in a vacuum evaporator for nanoLC-MS/MS analysis. NanoLC-MS/MS analyses were performed by using LTQ-Orbitrap (Thermo Fisher Scientific, Rockwell, IL, USA). Peptides and proteins were identified by means of automated database searching using Mascot version 2.3 (Matrix Science, London, UK) against the TAIR database (release 10) with a precursor mass tolerance of 3 ppm, a fragment ion mass tolerance of 0.8 Da, and strict trypsin specificity allowing for up to two missed cleavages. Cysteine carbamidomethylation was set as a fixed modification, and methionine oxidation and phosphorylation of serine, threonine, and tyrosine were allowed as variable modifications. Peptides were considered to be identified if the Mascot score was over the 95% confidence limit based on "identity" score of each peptide and if at least three successive y or b ions with an additional two or more y, b ions were observed. False-positive rate was evaluated by searching against a randomized decoy database created by the Mascot Perl script, and estimated at less than 1% for phosphopeptide identification in all LC-MS data. Phosphorylation sites were unambiguously determined when b or y ions were between which the phosphorylated residue exists

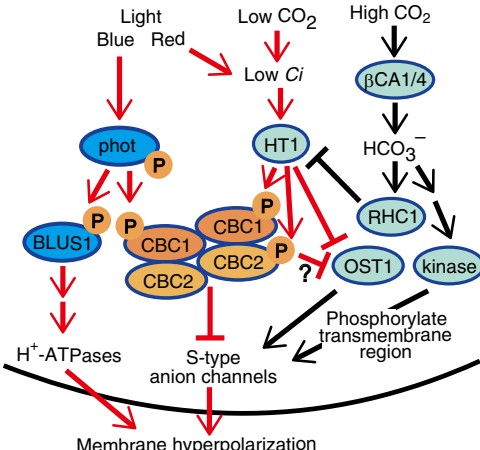

**Fig. 7** CBCs function where signals from BL and CO$_2$ converge. Phototropins that perceive BL undergo autophosphorylation and finally activate an H$^+$-ATPase via phosphorylation of BLUS1 kinase. Phototropins inhibit the S-type anion channel activities via CBC1 and CBC2 in response to BL. phot1 interacts with both CBC1 and CBC2 and phosphorylates CBC1. CBC1 interacts with both CBC1 and CBC2. CBC2 interacts with CBC2. CBC1 and CBC2 are located in the same pathway as HT1. Both CBC1 and CBC2 are required for stomatal opening in response to low concentrations of CO$_2$. CBC1 and CBC2 interact with and are phosphorylated by HT1. Carbonic anhydrases βCA1 and βCA4 produce HCO$_3^-$ from CO$_2$; the resulting HCO$_3^-$ activates RHC1, and the RHC1 inhibits HT1 kinase. The inhibitory effect of HT1 on OST1 is released in response to high concentrations of CO$_2$. The activated OST1 activates the S-type channels of SLAC1, thereby, causing stomatal closure. There are protein kinase pathways other than OST1 that lead to stomatal closure in response to high CO$_2$. Substrates for CBCs are unknown. Red lines signify stimulated stomatal opening and black lines signify accelerated stomatal closure

were observed in the peak list of fragment ions. Based on peptide information (observed *m/z* of monoisotopic ion and retention time) obtained by database searching, the LC-MS peak area of each peptide in all samples was determined by integration of ion intensity in survey MS scan. The peak integration was performed by Gaussian approximation of an extracted ion chromatogram within 5 mDa of the observed *m/z* using Mass Navigator v1.2.

We provided a supplemental excel file of all peptides identified, their MASCOT score, precursor charge, and *m/z*, peptide identification score in full PPlist as Supplementary Data 1. The MS data files are available from ProteomeXchange with the identifier PXD006586.

**Generation of transgenic plants**. Plants were transformed using *Agrobacterium tumefaciens*. To generate restoration lines, genomic fragments, including the promoter and coding region, were ligated to the pBluescript II SK (+) vector (Stratagene). The 3′-UTR region was introduced into the *Bam*HI and *Sal*I sites of the pCAMBIA1300 vector (Cambia), and then sGFP (GFP) was cloned into the *Bam*HI site. Subcloned regions were amplified and inserted into the same vector at the *Sma*I site. These cloning steps were performed using the In-Fusion system (Clontech); the primers used for CBC1 and CBC2 cloning are listed in Supplementary Table 4.

**Site-directed mutagenesis**. We constructed mutated genes of CBC1 by QuikChange Site-Directed Mutagenesis Kit (Stratagene), and listed the primers used for cloning of CBC1 (S43A S45A)-GFP in Supplementary Table 4.

**Generation of specific antibodies against CBC1 and CBC2**. The polyclonal antibodies against the GST-tagged recombinant protein fragments of the respective CBC1 and CBC2 (CBC1: Asp$^{25}$ to Ile$^{103}$, CBC2: Gln$^{22}$ to Ile$^{77}$) were raised in rabbits. The fragments were generated in *E. coli* as antigens, and were purified by glutathione-Sepharose beads with elution by reduced glutathione.

**Gas exchange and thermal imaging**. Stomatal conductance was determined by gas exchange using a Portable Photosynthesis System (LI-6400, LI-COR). Leaf temperature was measured by infrared thermography (TVS-8500, NEC Avio

Infrared Technologies) and visualized[21]. *Arabidopsis thaliana* plants were kept in the dark before the measurements.

**Measurement of stomatal aperture in epidermal peels**. Epidermal peels of 4–6-week-old plants were detached from dark-adapted *Arabidopsis* leaves. Peels floating on basal reaction mixture that contained 5 mM MES-bistrispropane (pH 6.5), 50 mM KCl, and 0.1 mM CaCl$_2$ were illuminated with RL (50 µmol m$^{-2}$ s$^{-1}$) and BL (10 µmol m$^{-2}$ s$^{-1}$) superimposed on the RL for 2 h to open stomata. The epidermal peels were collected on a 58 µm nylon mesh, rinsed with distilled water, and the stomatal apertures were measured microscopically. In Fig. 2a, b, once stomata opened in response to light, the peels were transferred to 5 mM MES-bistrispropane (pH 6.5) buffer containing 50 mM KCl or without KCl (control) and stomatal apertures were measured at the indicated times. To test the effect of an anion channel blocker on stomatal movement, anthracene-9-carboxylic acid (9-AC) (Sigma) dissolved in DMSO was added to the epidermal peels at a final concentration of 100 µM[59]. ABA-induced stomatal closure was conducted according to the method of ref. [42] with modification. To preopen stomata, leaves from 4- to 6-week-old plants were floated on opening buffer containing 5 mM KCl, 50 µM CaCl$_2$, and 10 mM MES-Tris (pH 6.15) in the light (80 µmol m$^{-2}$ s$^{-1}$). Then, epidermal peels were detached from the leaves and immersed in opening buffer in the presence of ABA. Stomatal apertures were measured after illuminating the epidermal peels with white light (80 µmol m$^{-2}$ s$^{-1}$) for 2 h.

**GCPs and H$^+$ pumping**. GCPs were prepared enzymatically unless otherwise noted[21]. BL- and Fc-dependent H$^+$ pumping were measured with a glass pH electrode[21].

**Patch clamping**. For patch clamping, GCPs were prepared according to the method of ref. [68]. The bath solution contained 30 mM CsCl, 2 mM MgCl$_2$, 1 mM CaCl$_2$, and 10 mM MES-Tris (pH 5.6). The osmolality of the bath solution was adjusted to 485 mmol l$^{-1}$ using D-sorbitol. The pipette solution contained 150 mM CsCl, 2 mM MgCl$_2$, 6.7 mM EGTA, 5 mM MgATP, 10 mM HEPES-Tris (pH 7.1), and sufficient CaCl$_2$ to result in a final concentration of 2 µM free Ca$^{2+}$. The osmolality of the pipette solution was adjusted to 500 mmol l$^{-1}$ using D-sorbitol. GCPs were pre-incubated in bath solution containing 1.2 mM NaHCO$_3$ for 5–20 min[43] under RL (700 µmol m$^{-2}$ s$^{-1}$) or RL (600 µmol m$^{-2}$ s$^{-1}$) plus BL (100 µmol m$^{-2}$ s$^{-1}$). Free concentrations of bicarbonate and CO$_2$ were calculated using the Henderson–Hasselbalch equation[44, 46]. The recording chamber was continuously perfused with bath solution. The S-type anion channel currents in GCPs were recorded by the whole-cell patch-clamp technique[43, 69]. Whole-cell recording of the inward K$^+$ current ($I_{Kin}$) was performed[57].

**Phosphorylation assays**. Phosphorylation of phototropins, BLUS1, CBC1, and CBC2 was terminated 1 min after the start of BL illumination by adding TCA to GCPs. Phosphorylation of these three proteins was determined by a mobility shift assay and phosphorylation of H$^+$-ATPase was estimated from a protein blot using 14-3-3 protein[19]. The amount of these proteins was estimated by immunoblot analysis[21]. Uncropped immunoblots are shown in Supplementary Fig. 10.

**CBC1 and CBC2 phosphorylation by Phos-tag PAGE**. GCP proteins were subjected to Phos-tag PAGE according to the manufacturer's instructions (Wako). For the separation of CBC proteins, 20 µM Phos-tag and 40 µM MnCl$_2$ for CBC1 and 40 µM Phos-tag and 80 µM MnCl$_2$ for CBC2 were used. The resolving gel contained 10 or 6% acrylamide/bis mixed solution (37.5:1) (Wako). Before blotting, the gels were washed twice in Towbin buffer containing 5 mM EDTA for 10 min, and washed again with Towbin buffer for 10 min. Uncropped immunoblots are shown in Supplementary Fig. 10.

**Protein expression and in vitro kinase assay**. For expression of GST-tagged protein, CBC1, CBC1 (D271N), CBC2, CBC2 (D245N), HT, HT1 (D184N), and HT1 (P240L) were cloned into pCold$^{TM}$GST DNA (TAKARA). P1C and P1C (D806N) were cloned into the pGEX-2T vector (GE Healthcare)[61]. All of the GST-tagged proteins were expressed in *E. coli* strain Rosetta$^{TM}$2 (DE3) (Novagen). The cells were grown at 37 °C in Luria–Bertani (LB) broth containing 50 mg ml$^{-1}$ ampicillin until the OD$_{600}$ reached 0.4–0.6. Isopropyl-β-D-thiogalactopyranoside was added to a final concentration of 0.2 mM, and the culture was incubated at 15 °C overnight. The cells were collected and resuspended in 20 mM Tris-HCl (pH 7.4) and 140 mM NaCl and were disrupted by sonication in the presence of 0.1% (v/v) Triton X-100. The disrupted cells were centrifuged, and the resulting supernatant was incubated with glutathione-Sepharose 4B (GE Healthcare) for 1 h at 4 °C. The glutathione-Sepharose beads were washed three times with 20 mM Tris-HCl (pH 7.4), 140 mM NaCl, and 0.1% Triton X-100, and once with 50 mM Tris-HCl buffer (pH 8.0). The GST fusion proteins were eluted with 10 mM reduced glutathione in 50 mM Tris-HCl buffer (pH 8.0). Purified proteins were stored in 50% (v/v) glycerol at −20 °C. In vitro phosphorylation assays were performed in a reaction mixture of 30 µl that contained 50 mM Tris-HCl (pH 7.5), 10 mM MgCl$_2$, 3.3 µM ATP, 20 µCi of [γ-$^{32}$P] ATP (3000 Ci mmol$^{-1}$, PerkinElmer), and purified proteins. Artificial substrates of casein (casein (dephosphorylated from

bovine milk), Sigma), MBP (MyBP, Sigma), and histone (histone (type III-S from calf thymus), Sigma) of 5 µg were used. The reaction proceeded for 2 h at 15 °C was stopped by adding SDS sample buffer and then boiled at 95 °C for 3 min. The samples were subjected to SDS-PAGE, and the CBB-stained gels were dried. Phosphorylated proteins were identified by autoradiography with an image plate and analyzed with an FLA5100 Phosphoimager (Fujifilm). Uncropped immunoblots are shown in Supplementary Fig. 10.

**In vitro pull-down assays**. For in vitro pull-down assays, His-tagged and FLAG-tagged proteins were synthesized[21]. Synthesized proteins (50–100 µl) were mixed with 450–900 µl binding buffer containing 20 mM Tris-HCl (pH 7.4), 140 mM NaCl, and 0.1% (v/v) Triton X-100, and the mixtures were kept on ice for 10 min. After centrifugation at 12,000×g for 10 min at 4 °C, the supernatants were mixed with anti-FLAG M2 Affinity Gel (Sigma-Aldrich) for 1 h at 4 °C. After washing the gel three times, the bound proteins were eluted and subjected to immunoblotting using THE™His Tag antibody, mAb, mouse (1:4000, GenScript, #A00186-100), and mouse TrueBlot® ULTRA: anti-mouse Ig HRP (1:1000, Rockland Immunochemicals, # 18-8817-30). FLAG-tagged proteins were detected by a monoclonal anti-Flag M2-peroxidase (HRP) antibody produced in a mouse (1:1000, Sigma-Aldrich, #A8592). Uncropped immunoblots are shown in Supplementary Fig. 10.

**BiFC experiments**. For BiFC assays, nYFP- and cYFP-fused constructs were co-transformed into mesophyll protoplasts[70] by PEG-calcium transfection[71]. To generate nYFP- and cYFP-fused constructs, expression cassettes of pSPYNE173 and pSPYCE (M) were introduced into the pRI101 vector. CDS fragments of *PHOT1*, *CBC1*, *CBC2*, and *HT1* were introduced into the *Bam*HI and *Sal*I sites of the vector. Fluorescent images were obtained using a confocal laser scanning microscope (Digital Eclipse C1; Nikon). Wavelengths for excitation and emission of YFP were 488 and 515–530 nm, and those for chlorophyll fluorescence were 543 and 590 nm long-pass.

**Data availability**. MS data files have been deposited at ProteomeXchange with the identifier PXD006586. The authors declare that all other data supporting the findings of this study are available within the manuscript and its supplementary files or are available from the corresponding author on request.

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

## Acknowledgements

We are grateful to Dr M. Doi and Dr Y. Takahashi for their suggestions. We thank M. Aibe, E. Abe, and Dr E. Gotoh for their technical assistance. We are indebted to Dr H. Tachida for the discussion on the evolutionary aspect. We are also grateful to the Salk Institute Genomic Analysis Laboratory, the NASC, and the Syngenta Arabidopsis Insertion Library collection for providing the seeds. This work was supported by JSPS KAKENHI Grant numbers, 21227001, 26251032 (to K.S.), 26711019, and 15K14552 (to A.T.), and MEXT KAKENHI Grant numbers, 23120521 and 25120719 (to A.T.), and Grant-in-Aid for JSPS fellows Grant number 13J05118 (to A.H.).

## Author contributions

A.H., A.T., and K.S. conceived and designed the experiments. A.H. performed most of the experiments. N.S. did the mass spectrometric analysis. S.M., E.O., and Y.M. did the patch-clamp experiments and analysis. Y.T. assisted in in vitro transcription/translation experiments. A.H. and K.S. wrote the manuscript.

## Additional information

**Competing interests:** The authors declare no competing financial interests.

