## [Peer Review File · Nature Communications]

Reviewers' comments:

Reviewer #1 (Remarks to the Author):

The work by Hiyama et al characterizes the functions of two protein kinases, CBC1 and CBC2, as negatively regulating stomatal conductance and S-type anion channel activity in response to blue light and low CO₂ by utilizing genetic, biochemical, physiological and electrophysiological assays in Arabidopsis. Furthermore, the interaction between CBC1/2 and HT1, and phototropins was studied in vitro. This study provides important new insights into the convergence of CO₂ and blue light signaling pathway in regulating stomatal movements.

The work is carried out well and is sound. Some more information should be added to render the paper easier to follow and some points in the study are still not clear and should be answered.

Specific comments:

1. In figure 1, the present data show that the *cbc1/2* double mutant has higher leaf temperature than WT plants in response to blue light. What about leaf temperature of the double mutant in the absence of blue light?
2. In figure 2, the authors performed stomatal opening assays indirectly suggesting that CBC1 and CBC2 negatively regulates SLAC1 and SLAH3 activity during stomatal opening. What about the stomatal closure process in this *cbc* double mutant compared to WT, since SLAC1 and SLAH3 function in stomatal closure?
3. The experimental error in figure 2 is relatively large, which could be expected. The authors should more carefully interpret these data and provide some statistical information in the results section and figure captions and include the number of experiments. The experimental error is not a major problem here, since the much more robust stomatal conductance measurements show very strong *cbc* phenotypes. However, the interpretation that SLAC1 and SLAH3 might be regulated should be stated much more carefully.
4. The presented data are relatively clear suggesting that *cbc1/2* double mutant plants are insensitive to both blue light and low CO₂. It will be very helpful if the authors could add gas-exchange experiments in Arabidopsis wild-type and *cbc1/2* mutant plants to test stomatal opening in the presence of CO₂ level shifts with the presence of blue light treatment.
5. The authors performed whole cell patch clamp experiments by incubating the guard cells with bicarbonate in the bath solution. It may be helpful if the authors also add patch clamp experiments by adding bicarbonate in the intracellular solution as also included in the cited methods of Hu et al., (2010). This may not be essential in this paper, but see the next comment 6.
6. In figure 3, if the total numbers of patch clamped guard cells are shown in the figure legend. The numbers seem to be slightly limited. Adding data with intracellular HCO₃⁻ or more repeats with extracellular HCO₃⁻ could address this comment.
7. Statistical analyses are missing for the data performed in Fig. 1g and Fig. 3. Even though statistical analyses were done in Fig. 2 and 5, they are poorly described in the figure legends. The authors should add more information.
8. BiFC experiments are lacking proper controls. Fluorescence intensities should be compared to positive and true negative controls.

Minor Comments:

9. In figure 3, it seems like bicarbonate still can activate S-type channel currents in *cbc1/2* mutant plant guard cells. However, Fig. 5a shows that *cbc1/2* mutant plants did not show response to high CO₂ (800 ppm) in stomatal closing period. Please explain. The stomata are already apparently closed in Fig. 5. If the authors think this explains the data they could add text.
10. On page 6, second part, Fig. 1c shows the detail information of *cbc* T-DNA insertion mutants rather than Fig. 1d

11. In Fig. 2a,b, there are "dark" and "light" treatments. However, in Fig. 2c,d,e, there are "dark" and "RL+BL" treatments. Are light and RL+BL the same? If not, describe clearly in the results and Figure legend.
12. Reference 57 was published in 2002 rather than 2012.
13. Page 8: Earlier research showing that 50% K⁺ current reduction is not sufficient should also be discussed.
14. In the abstract and other parts of the manuscript the authors should refer to S-type channels rather than SLAC1/SLAH3, since S-type channel regulation was studied but not yet SLAC1/SLAH3 regulation directly here.

In conclusion this is an interesting and important manuscript that reports relevant new findings that are highly suitable for Nature Communications and will be of interest to the community.

Reviewer #2 (Remarks to the Author):

Understanding the molecular grounds of the way stomata open in response to light and low CO₂ concentrations, in order to maximize photosynthesis, is of prime importance for the breeding of high yield crops.

In this manuscript the authors report on a search for proteins that are phosphorylated in response to BL via phototropin receptors.

They identified a MAP kinase, which they named CONVERGENCE of BLUE LIGHT and CO₂ (CBC1). CBC1 is rapidly phosphorylated in BL, through a signaling pathway that depends on PHOT-receptors. Stomata of that have lost function of CBC1 and the close homolog CBC2 are impaired in BL-, red light, and low CO₂ induced stomatal opening.

In contrast to their expectations, the CBC1 and 2 MAP kinases are not required for the BL-dependent activation of plasma membrane H⁺-ATPases, which is a major topic of interest in the Shimazaki Lab. However, the authors found instead that the CBC-type protein kinases mediate the BL-induced inhibition of the S-type anion channels in guard cells.

The authors finding of CBCs is interesting and important to understand guard cell signaling, but there still remain open questions (listed below) as to role of the MAP kinases in integrating Light and CO₂ signaling that need to be answered.

Comments and questions:

1. The authors state that CBC1 and CBC2 genes are expressed in guard cells, but in other cell types as well (Supplementary Fig. 3).

Comment: For the reader to better follow the latter guard cell-based argumentation needs to find GUS and/or GFP marker based localization side by side with qPCR data.

2. Figure 1 h shows that in IRGA measurements both RL- and BL-dependent stomatal opening are inhibited to the same degree in the single and double cbc mutants

Comment 1: the cbc1/2 double mutant has only half of the stomatal conductance at the start of the IRGA-measurements (Fig. 1h), but this difference is not observed in epidermal strip assays (Fig. 1g and SFig. 4f). The authors should explain why the experiments with epidermal strips do not match those of stomata in the intact plant, in this respect.

Comment 2: The data obtained with epidermal strips differ in absolute and relative amplitude to the IRGA measurement. Thus the reader would like to see IRGA data side-by-side with strip data.

3. In response to BL inward K⁺ currents were reduced approximately 50% in the double cbc mutant (Supplementary Fig. 4e). The authors, however, state that such reduction in K⁺ current will not affect

stomatal opening.

Comment: Even though the author agrees with this conclusion, it should be explained in more detail, as several published papers have suggested that a 50% reduction in inward K⁺ currents of guard cells will influence stomatal opening (Schwartz et al., 1994; Armstrong et al., 1995). Experiments with inward K⁺ channel mutants have shown that loss of the major K⁺ channel KAT1 does not affect stomatal movements (Szyroki et al., 2001), but only if the all K⁺ channels light-induced stomatal opening is impaired (Lebaudy et al., 2008).

4. The authors suspected that the S-type anion channels in the mutant might release more Cl⁻ from the guard cells into the medium. This hypothesis was tested with the non-specific anion channel pharmacon 9-AC.

Comment: to confirm their finding the authors by more direct evidence via e.g. proper EDXA analysis of K and Cl relative content.

5. The authors found apertures were smaller in the *cbc1 cbc2 slac1-4* triple mutant than in *slac1-4*. This result is probably due to the activation of SLAH3 in the mutant. Likewise, stomatal apertures were smaller in *cbc1 cbc2 slah3-3* than in *slah3-3* (Fig. 2d), probably due to activated SLAC1 in the triple mutant.

Comment: this postulate can be tested by guard cell patch clamping using Cl⁻ vs NO₃⁻ to distinguish between the contribution of SLAC1 and SLAH3.

6. WT and *cbc1 cbc2* Guard Cell Protoplasts showed no large anion currents under control conditions, as has been reported (Ref.47,50). The activation of anion currents by bicarbonate was observed in GCPs from the *cbc1 cbc2* double mutant (Fig. 3a, c). However, in contrast to WT GCPs, the enhanced current of *cbc1 cbc2* GCPs was not suppressed by BL.

Comment 1: what about RL? Does RL inhibit calcium-induced S-type anion currents in GCPs? Mori et al., 2006 and Geiger et al., 2010 showed macroscopic anion currents in the presence of 2 μM cytosolic free calcium. The authors need to show these calcium-induced S-type anion currents in the absence of RL as a control.

Since several authors have spent time in the Schroeder lab and co-authored S-type anion channel studies, they should be aware of the fact that 2 μM cytosolic free calcium is sufficient to activate S-type anion channels in GCPs and that these controls need to be shown in the absence of RL.

Comment 2: The reader needs to know whether the calcium and ABA activation (without RL) of S-type channels (Schroeder and Hagiwara, 1989; Chen et al., 2010; Stange et al., 2010) in *cbc* mutants was WT like too?

7. Phototropism directly phosphorylated CBC1 in vitro

Comment: Does BL lead to phosphorylation of SLAC1 and SLAH3 (Demir et al., 2013) in guard cells? Do phototropins directly phosphorylate the SLAC1 and SLAH3 channels?

8. The authors claim that CBCs function downstream of HT1 and ultimately inhibit the S-type anion channels.

Comment: This hypothesis can be tested in the way J. Schroeder's lab did for the ABA-signaling cascade (Brandt et al., 2012).

9. The authors argue that it is possible that CBCs inhibit OST1 by phosphorylation, thereby suppressing S-type anion channels and/or protein kinases other than OST1 and causes CO₂-dependent stomatal closure by activation of SLAC1 through phosphorylation of the transmembrane region.

Comment: To answer this question the authors should include in their studies the OST1 mutant as well as CPK mutants.

10. Stomata in the *cbc1 cbc2* double mutant were clearly sensitive to ABA in epidermal peels, as has been reported for the *ht1* mutants and the authors claim CBCs do not play a role in ABA-induced stomatal closure.

SLAC1 and SLAH3 are activated in response to ABA and essential for ABA-dependent stomatal closure. CBCs are supposed to inhibit S-type channel opening, but the double mutant has a WT ABA response. Comment: Shouldn't at least the *cbc1 ox* show an ABA insensitive phenotype? The reader would like to find these data?

11. S-type channels do not conduct malate. QUAC1 in Arabidopsis is required for malate induced stomatal closure. Malate is formed by PEP and CO₂.

Comment: Does the loss of *cbc1* and *2* affect QUAC1 activity?

12. SFIG. 4(d) BL- and Fc-dependent H⁺-pumping from GCPs and (f) Fc-induced stomatal opening in epidermal peels of WT and *cbc1 cbc2*.

Comment: These data lack statistics, which should be provided.

13. Figure 2. The title of this figure is: "Enhancement of S-type anion channel activity in *cbc1 cbc2* mutants." Note, that title is misleading, as changes in anion channel activity are only shown in Fig. 3.

14. Since the authors did not use heterologous expression system such as *Xenopus* oocytes, they should in the discussion mention that future reconstitution studies would be necessary to underpin their findings.

Armstrong, F., Leung, J., Grabov, A., Brearley, J., Giraudat, J., and Blatt, M.R. (1995). Sensitivity to abscisic acid of guard-cell K⁺ channels is suppressed by *ABI1-1*, a mutant Arabidopsis gene encoding a putative protein phosphatase. *Proc. Natl. Acad. Sci. U. S. A.* 92, 9520-9524.

Brandt, B., Brodsky, D.E., Xue, S., Negi, J., Iba, K., Kangasjarvi, J., Ghassemian, M., Stephan, A.B., Hu, H.H., and Schroeder, J.I. (2012). Reconstitution of abscisic acid activation of SLAC1 anion channel by CPK6 and OST1 kinases and branched *ABI1* PP2C phosphatase action *Proc. Natl. Acad. Sci. U. S. A.* 109, 10593-10598

Chen, Z.H., Hills, A., Lim, C.K., and Blatt, M.R. (2010). Dynamic regulation of guard cell anion channels by cytosolic free Ca²⁺ concentration and protein phosphorylation. *Plant J.* 61, 816-825.

Demir, F., Horntrich, C., Blachutzik, J.O., Scherzer, S., Reinders, Y., Kierszniowska, S., Schulze, W.X., Harms, G.S., Hedrich, R., Geiger, D., and Kreuzer, I. (2013). Arabidopsis nanodomain-delimited ABA signaling pathway regulates the anion channel SLAH3. *Proc. Natl. Acad. Sci. U. S. A.* 110, 8296-8301.

Lebaudy, A., Vavasseur, A., Hossy, E., Dreyer, I., Leonhardt, N., Thibaud, J.B., Very, A.A., Simonneau, T., and Sentenac, H. (2008). Plant adaptation to fluctuating environment and biomass production are strongly dependent on guard cell potassium channels. *Proc. Natl. Acad. Sci. U. S. A.* 105, 5271-5276.

Schroeder, J.I., and Hagiwara, S. (1989). Cytosolic calcium regulates ion channels in the plasma-membrane of *Vicia faba* guard-cells. *Nature* 338, 427-430.

Schwartz, A., Wu, W.H., Tucker, E.B., and Assmann, S.M. (1994). Inhibition of inward K⁺ channels and stomatal response by abscisic acid - an intracellular locus of phytohormone action. *Proc. Natl. Acad. Sci. U. S. A.* 91, 4019-4023.

Stange, A., Hedrich, R., and Roelfsema, M.R.G. (2010). Ca²⁺-dependent activation of guard cell anion channels, triggered by hyperpolarization, is promoted by prolonged depolarization. *Plant J.* 62, 265-276.

Szyroki, A., Ivashikina, N., Dietrich, P., Roelfsema, M.R.G., Ache, P., Reintanz, B., Deeken, R., Godde, M., Felle, H., Steinmeyer, R., Palme, K., and Hedrich, R. (2001). *KAT1* is not essential for stomatal opening. *Proc. Natl. Acad. Sci. U. S. A.* 98, 2917-2921.

Reviewer #3 (Remarks to the Author):

Blue light and CO₂ signals converge to regulate light-induced stomatal opening.

Review: Guard cells are endowed with sensor systems that allow them to fine-tune stomatal apertures in response to changes in light intensity, carbon dioxide concentration, temperature, and drought. This fine tuning serves to keep the optimal balance of photosynthesis with transpiration for a given set of environmental conditions and protect against excessive water loss under drought in high temperature conditions. It is well known that blue light induces stomatal opening by the activation of the phototropins and that high carbon dioxide induces stomatal closure. Conversely low carbon dioxide induces stomatal opening. Although there is a very large and sophisticated literature derived from electrophysiological studies on the channels and ion pumps involved, There remain serious gaps in knowledge of the signal-transduction pathways functioning with changes in carbon dioxide concentration and with phototropin activation.

Hiyama et al. characterize a pair of kinases, CBC1 and CBC2 that negatively regulate the two slow anion channels SLAC1 and SLAH1 in response either to low carbon dioxide or to blue light. The function of these channels is essential for stomatal closing. A *cbc1 cbc2* double mutant was severely impaired in stomatal opening. In vitro, CBC1 was shown to be phosphorylated by an active phot1 preparation (both in-vitro pull-down assays and in vivo BiFC indicated a direct association of CBC1 with phot1. Curiously, the guard-cell pathway involving the phototropins, BLUS1, and a proton ATPase was unimpaired. A mix of electrophysiological, genetic, and physiological experiments with leaves/epidermal peels strongly implicated blue-light-activated CBC1 and CBC2 in negatively regulating SLAC1 and SLAH1 and consequently inhibiting stomatal closure.

HT1 is another kinase known to be a negative regulator of carbon dioxide-induced stomatal closure, the author used genetic experiments with *cbc1*, *cbc2*, and *ht1* mutants to show that they functioned on the same pathway to inhibit stomatal closing. They were then able to show both by in vitro pull-down assay and in vivo BiFC that these proteins interacted directly. Thus either light-activated phototropins, or low carbon dioxide-activated HT1 phosphorylated CBC1 and induce association with CBC2. The results are the first demonstration of components shared between these two pathways.

I find this study a really solid advance in our knowledge of stomatal regulation by two different environmental parameters, establishing a common pathway, and recommend publication. I do have some minor comments however:

1. HT1 needs a little more explanation in the abstract.
2. P. 6, par. 1, l. ...H⁺-ATPases, respectively, reach...
3. P7. Par. 2. L. 2: ...but substantially to blue light.. (delete and largely)
4. P. 9, l. 2: "We hypothesize that the gene disruption..." L. 4: To test this hypothesis directly..." As written, text suggests author bias toward a favorable result.
5. In two places, "pull-downed" should be replaced with "pulled down"
6. P. 13, par. 2, last line: use "relevance" here not "meaning"
7. Fig. 1 legend, epidermises, not epidermis. The images in 1f were not easy to see. Fig. 1g, why standard deviation? What differences are significant?
8. Fig. 1 legend, last line: A sample of four is insufficient to calculate a meaningful standard error.
9. Figures 4d and 6a: These BiFC images merely establish that there is interaction in the periphery of guard-cell protoplasts, not that the interacting components are "in the plasma membrane."

Reviewer #4 (Remarks to the Author):

The manuscript "Blue light and CO₂ signals converge to regulate light-induced stomatal opening", submitted by Hiyama et al. describes two new kinases that link blue light and CO₂ induced stomatal opening. The results are original and are of high interest to the scientific community. The mechanism of activation of CBC1 and CBC2 and their regulation of S-type anion channels in response to both signaling pathways need however more work to support the claims presented by the authors. In detail these questions should be addressed prior to resubmission:

Phosphorylation data

A potential mode of CBC regulation might be the phosphorylation of CBC1 or potentially CBC1/CBC2.

- 1) The authors do not mention how many biological replicates were performed for the phosphoproteomic study to quantify CBC1 phosphorylation.
- 2) No protein database, settings or statistical data are given for the quantitation of phosphorylation changes as well as for the peptide identification.
- 3) Several ions, including y11 and y14 are not marked in the graph in Figure1a.
- 4) The authors could mention that the identified sites in CBC1 have been published before (see PhosPHAT 4.0)
- 5) Was there only a doubly phosphorylated peptide of CBC1 identified or did the singly phosphorylated peptides not change in abundance?
- 6) Did the author quantify nonphosphorylated peptides of CBC1 to rule out changes in protein abundance?
- 7) The raw data should be deposited in a common proteomics dataset, e.g. PRIDE, so that others, including the reviewers, can confirm the results.
- 8) In the figure caption of figure 1c, it says schematic structures indicate phosphorylation sites of CBC1 and CBC2. I do not find those in the actual figure.

Thermal imaging

- 1) The main text claims that the cbc1 mutant is impaired in response to BL. This is very hard to judge based on Figure1d. Thermal imaging should be quantified with multiple replicates and statistically analyzed to support those claims.

Stomata assays

- 1) The authors claim that BL dependent stomatal opening was slightly impaired in the single mutants of cbc1 and cbc2. Without a statistical analysis this cannot be judged from Figure1g. I might also have misunderstood the statement in the introduction that RL induced stomatal opening is not apparent in epidermal peels of Arabidopsis. Because why otherwise would the authors measure RL induced stomatal opening in Figure1g?
- 2) Stomatal conductance measurements in Figure1h need a statistical analysis. Normalization to the same starting conductance might also be a good idea to claim RL and BL effects.
- 3) Figure2a and Figure2b contain asterisks, indicating some kind of statistical test, which is not mentioned in the figure caption or main text. Please include a statistical analysis.
- 4) As a general note, if statistical analyses of stomata assays are performed, keep in mind that stomatal opening and closing does not follow a normal distribution, care should be taken in selecting a statistical test.
- 5) The authors claim that stomatal apertures were smaller in the slah3cbc1ccb2 triple mutant than in slah3, probably due to activation of SLAC1. Based on the findings presented in the manuscript, SLAH3 has no phenotype as compared to wildtype plants, and the triple mutant merely reflects the

phenotype of the cbc1cbc2 double mutant.

6) 1mM CO₂ appears quite excessive for stomata assays. I know it was used before, but if I am not mistaken, 1 mM CO₂ calculates to 22.000 ppm (http://hiq.linde-gas.com/en/specialty_gases/gas_concentration_un_its_converter.html). Comments would be appreciated.

Electrophysiology

1) No statistics on the electrophysiological assays.

Phosphorylation and kinase assays

Besides the phosphoproteomic data presentation, that lays the foundation of a putative phosphoregulation of CBC proteins, the kinase assays are the biggest problems of the manuscript. The authors do not show that Phototropin or HT1 phosphorylate CBC proteins at the site they identified in their screen. The presented experiments do not sufficiently support a regulation by Phototropin or HT1. Phosphomimic mutants, in vivo phosphorylation analyses in respective phototropin or ht1 mutant lines might help to further substantiate such claim.

1) The Phostag shift in Figure4a requires an additional phosphatase control to confirm this shift is due to phosphorylation.

2) CBC2 does show multiple bands in Figure4a. These might be different phosphorylation forms. The second one from the bottom might even increase in response to BL. Could the authors comment.

3) Figure4b is not convincing and minimum requires a phosphatase treatment.

4) The kinase assay in Figure1e contains more enzyme (P1C) than substrate (CBC). Under these conditions nearly every Ser/Thr kinase can phosphorylate CBC.

5) In general, since the authors are able to quantify phosphorylation changes in planta. Why don't they do this for ht1 and phot mutants.

Reviewers' comments:

Reviewer #1 (Remarks to the Author):

The work by Hiyama et al characterizes the functions of two protein kinases, CBC1 and CBC2, as negatively regulating stomatal conductance and S-type anion channel activity in response to blue light and low CO₂ by utilizing genetic, biochemical, physiological and electrophysiological assays in Arabidopsis. Furthermore, the interaction between CBC1/2 and HT1, and phototropins was studied in vitro. This study provides important new insights into the convergence of CO₂ and blue light signaling pathway in regulating stomatal movements. The work is carried out well and is sound. Some more information should be added to render the paper easier to follow and some points in the study are still not clear and should be answered.

Specific comments:

1. In figure 1, the present data show that the *cbc1/2* double mutant has higher leaf temperature than WT plants in response to blue light. What about leaf temperature of the double mutant in the absence of blue light?

We measured the leaf temperature 30 and 50 min after red light illumination in WT and the *cbc1 cbc2* double mutant. In both times, the leaf temperatures were higher by 0.5 °C in the double mutant than in WT (See below).

Plant materials	Leaf temperature (°C)		ΔLeaf temperature (°C)
	30 min	50 min	BL - RL
Wild type	23.9 ± 0.4	24.2 ± 0.4	-0.56 ± 0.19
cbc1	23.9 ± 0.4	24.2 ± 0.3	-0.47 ± 0.19*
cbc2	24.2 ± 0.4 *	24.5 ± 0.4 *	-0.36 ± 0.18 *
cbc1 cbc2	24.4 ± 0.3 *	24.7 ± 0.3 *	-0.28 ± 0.15 *

Ave. ± s.e.
(n = 12, pooled from triplicate experiments)

2. In figure 2, the authors performed stomatal opening assays indirectly suggesting that CBC1 and CBC2 negatively regulates SLAC1 and SLAH3 activity during stomatal opening. What about the stomatal closure process in this *cbc* double mutant compared to WT, since SLAC1 and SLAH3 function in stomatal closure?

It is difficult to precisely measure the process of stomatal closure in epidermal peels because the open size of aperture in the *cbc* double mutant is small. The rate of stomatal closure should be faster in the *cbc* double mutant than in WT, but we could not obtain clear results. We further inspected closure process in intact leaves, but we could not see the significant difference between the *cbc* mutants and WT.

3. The experimental error in figure 2 is relatively large, which could be expected. The authors should more carefully interpret these data and provide some statistical information in the results section and figure captions and include the number of experiments. The experimental error is not a major problem here, since the much more robust stomatal conductance measurements show very strong *cbc* phenotypes. However, the interpretation that SLAC1 and SLAH3 might be regulated should be stated much more carefully.

We measured stomatal aperture for 25 stomata and the measurements were repeated three times in different occasions. We noted the statistical information in the legends of Figure 2.

We also carefully interpreted these data and improved the expression:

- 1) We replaced the words "SLAC1 and SLAH3" by "S-type" in abstract.
- 2) We changed the title from "SLAC1 and SLAH3" to "S-type anion channels" in the head of second paragraph for figure 2.
- 3) We softened the description of the role of SLAH3 in stomatal closure in our experimental conditions.

4. The presented data are relatively clear suggesting that *cbc1/2* double mutant plants are insensitive to both blue light and low CO₂. It will be very helpful if the authors could add gas-exchange experiments in *Arabidopsis* wild-type and *cbc1/2* mutant plants to test stomatal opening in the presence of CO₂ level shifts with the presence of blue light treatment.

This is a good suggestion. We determined the stomatal responses by shifting CO₂ levels from 350 to 100 ppm in the presence of RL (100 $\mu\text{mol m}^{-2} \text{s}^{-1}$) or RL (90 $\mu\text{mol m}^{-2} \text{s}^{-1}$) plus BL (10 $\mu\text{mol m}^{-2} \text{s}^{-1}$). We used moderate intensity of light to allow further stomatal opening in response to the low CO₂ concentrations. We obtained the clear results, showing that stomatal opening was much larger in WT than the *cbc1 cbc2* double mutant in the presence of light. The degree of opening in WT was two-fold larger in the presence of light than in the absence of light (Fig. 5d). BL was more effective for low CO₂-induced stomatal opening than RL (Supplementary Table 3).

5. The authors performed whole cell patch clamp experiments by incubating the guard cells with bicarbonate in the bath solution. It may be helpful if the authors also add patch clamp experiments by adding bicarbonate in the intracellular solution as also included in the cited methods of Hu et al., (2010). This may not be essential in this paper, but see the next comment 6.

Hu et al., (2010) showed that the addition of bicarbonate/CO₂ to either intracellular solution or extracellular solution elicits S-type anion currents in *Arabidopsis* GCPs. In Figure 3, we showed that in WT GCPs, BL inhibits S-type anion channel when the channel activation is elicited by “extracellular” bicarbonate. Under this condition, bicarbonate/CO₂ signaling activation and signal integration of bicarbonate/CO₂ and BL can be achieved in GCPs that have “INTACT” cytoplasm. By contrast, our preliminary experiments revealed that when bicarbonate was added in “intracellular” solution, BL illumination gives no clear effect on the S-type anion currents (data not shown).

6. In figure 3, if the total numbers of patch clamped guard cells are shown in the figure legend. The numbers seem to be slightly limited. Adding data with intracellular HCO₃⁻ or more repeats with extracellular HCO₃⁻ could address this comment.

We apologize for the misleading description. This does NOT mean the total numbers of patch clamped guard cells. We revised the figure legend and now show exact numbers of tested guard cells for each treatment group.

WT control (n=7)

WT bicarbonate/CO₂, RL (n=7)

WT bicarbonate/CO₂, RL+BL (n=6)

cbc1cbc2 control (n=8)

cbc1cbc2 bicarbonate/CO₂, RL (n=7)

cbc1cbc2 bicarbonate/CO₂ RL+BL (n=5)

We noted these data in legends of Figure 3.

7. Statistical analyses are missing for the data performed in Fig. 1g and Fig. 3. Even though statistical analyses were done in Fig. 2 and 5, they are poorly described in the figure legends. The authors should add more information.

Thank you for pointing this out. We did statistical analysis for the data in the figures. Different letter labels indicate significant differences ($p < 0.05$).

On Fig. 1g (new Fig.1h); Data were presented in Fig.1h legend.

On Fig. 2a, b; We noted the meaning of asterisks in Figure legend.

On Fig. 3; We added labels. Different letter labels indicate significant differences ($p < 0.05$). We noted these in the legends of Fig. 3.

8. BiFC experiments are lacking proper controls. Fluorescence intensities should be compared to positive and true negative controls.

We think that most of the experiments used half YFP protein (nYFP) as negative controls shown in Fig. 4d and Fig. 6a as has been reported (Nature Comm. Tian et al., 2015, Figure 2).

We interpreted that “true negative controls” are the proteins that have been demonstrated not to interact with phot1 or HT1 in BiFC assays. According to your comment, we looked for the “true negative proteins” for phot1 and HT1 as controls. We found that PIN1 protein is a candidate for phot1 (Zhao et al., 2013, Plant Physiol.), but could not find such proteins for HT1.

We then tested the interaction between phot1 and PIN1 by BiFC assay and presented the data in SFig. 8 as a negative control, but did not perform the same assay for HT1. In addition to this, we added the “positive controls” for both phot1 and HT1 using bZIP63 proteins (right side panels of Fig. 4d, 6a)

Minor Comments:

9. In figure 3, it seems like bicarbonate still can active S-type channel currents in *cbc1/2* mutant plant guard cells. However, Fig. 5a shows that *cbc1/2* mutant plants did not show response to high CO₂ (800 ppm) in stomatal closing period. Please explain. The stomata are already apparently closed in Fig. 5. If the authors think this explains the data they could add text.

We cannot explain the results at present time. I think the same thing as suggested by reviewer. The discrepancy between stomatal response and the anion channel activity was also found in the *ht1* mutant (Xue et al., 2011). One of the reasons for this is that the stomata are already closed in the double mutant. In accord with this interpretation, stomata exhibited the smaller apertures in the *cbc1 cbc2* double mutant than those in WT under the dark (Fig.1g, Fig. 2c-e). We noted this in the text and presented the data (Supplementary Table1a).

10. On page 6, second part, Fig. 1c shows the detail information of *cbc* T-DNA insertion mutants rather than Fig. 1d

Thank you. We replaced Fig. 1d by Fig. 1c.

11. In Fig. 2a,b, there are “dark” and “light” treatments. However, in Fig. 2c,d,e, there are “dark” and “RL+BL” treatments. Are light and RL+BL the same? If not, describe clearly in the results and Figure legend.

Thank you very much for your suggestion. “Light” and “RL+BL” are the same. We replaced “Light” by “RL+BL” in this figure.

12. Reference 57 was published in 2002 rather than 2012.

Thank you. We corrected it.

13. Page 8: Earlier research showing that 50% K⁺in current reduction is not sufficient should also be discussed.

We discussed it in Discussion.

14. In the abstract and other parts of the manuscript the authors should refer to S-type channels rather than SLAC1/SLAH3, since S-type channel regulation was studied but not yet SLAC1/SLAH3 regulation directly here.

We replaced the words “SLAC1 and SLAH3” by “S-type”.

In conclusion this is an interesting and important manuscript that reports relevant new findings that are highly suitable for Nature Communications and will be of interest to the community.

Reviewer #2 (Remarks to the Author):

Understanding the molecular grounds of the way stomata open in response to light and low CO₂ concentrations, in order to maximize photosynthesis, is of prime importance for the breeding of high yield crops.

In this manuscript the authors report on a search for proteins that are phosphorylated in response to BL via phototropin receptors.

They identified a MAP kinase, which they named CONVERGENCE of BLUE LIGHT and CO₂ (CBC1). CBC1 is rapidly phosphorylated in BL, through a signaling pathway that depends on PHOT-receptors. Stomata of that have lost function of CBC1 and the close homolog CBC2 are impaired in BL-, red light, and low CO₂ induced stomatal opening.

In contrast to their expectations, the CBC1 and 2 MAP kinases are not required for the BL-dependent activation of plasma membrane H⁺-ATPases, which is a major topic of interest in the Shimazaki Lab. However, the authors found instead that the CBC-type protein kinases mediate the BL-induced inhibition of the S-type anion channels in guard cells.

The authors finding of CBCs is interesting and important to understand guard cell signaling, but there still remain open questions (listed below) as to role of the MAP kinases in integrating Light and CO₂ signaling that need to be answered.

Comments and questions:

1. The authors state that CBC1 and CBC2 genes are expressed in guard cells, but in other cell types as well (Supplementary Fig. 3).

Comment: For the reader to better follow the latter guard cell-based argumentation needs to find GUS and/or GFP marker based localization side by side with qPCR data.

We performed the experiment using GFP marker and presented the data. We confirmed that the both CBC1 and CBC2 were expressed in the cytosol of guard cells (SFig. 3b).

2. Figure 1 h shows that in IRGA measurements both RL- and BL-dependent stomatal opening are inhibited to the same degree in the single and double cbc mutants

Comment 1: the cbc1/2 double mutant has only half of the stomatal conductance at the start of the IRGA-measurements (Fig. 1h), but this difference is not

observed in epidermal strip assays (Fig. 1g and SFig. 4f). The authors should explain why the experiments with epidermal strips do not match those of stomata in the intact plant, in this respect.

In our experiences, we often observe such difference in stomatal opening between the epidermal peels and intact leaf. It is intrinsically difficult to quantitatively compare stomatal aperture in the epidermis and the conductance in intact leaves. For example, RL-induced stomatal opening is not evident in the epidermis of *Arabidopsis*, but the opening is clearly observed in the intact leaves as shown here. The difference is probably brought about by the situation in which the guard cells are placed: reduction of CO₂ concentration or transport of some unknown substances from mesophyll cells by light occurs around stomata in intact leaf, but such events do not occur in the epidermis.

Comment 2: The data obtained with epidermal strips differ in absolute and relative amplitude to the IRGA measurement. Thus the reader would like to see IRGA data side-by-side with strip data.

As noted above, it is not always adequate to compare the stomatal aperture in the epidermis with stomatal conductance in leaf. There may be unknown factors (ex: diffusible substances from mesophyll cells to guard cells in the leaf) affecting the stomatal responses in these plant materials. We think that it is not necessary to present the data side-by-side. However, we notice that the responses of stomata in the epidermis and the conductance in the leaf are qualitatively consistent throughout the experiments.

3. In response to BL inward K⁺ currents were reduced approximately 50% in the double cbc mutant (Supplementary Fig. 4e). The authors, however, state that such reduction in K⁺ current will not affect stomatal opening.

Comment: Even though the author agrees with this conclusion, it should be explained in more detail, as several published papers have suggested that a 50% reduction in inward K⁺ currents of guard cells will influence stomatal opening (Schwartz et al., 1994; Armstrong et al., 1995). Experiments with inward

K⁺ channel mutants have shown that loss of the major K⁺ channel KAT1 does not affect stomatal movements (Szyroki et al., 2001), but only if the all K⁺ channels light-induced stomatal opening is impaired (Lebaudy et al., 2008).

I am sorry that we misled the reviewer. Fig. S5e (former Fig. S4e) only indicated the capacity of voltage-dependent K⁺ current and were not directly related to BL response. The result suggested that the remaining capacity of K⁺ channels in the *cbc1 cbc2* double mutant is sufficient for stomatal opening. We responded to the related point in more detail in the reply for reviewer1, and discussed about it Discussion (comment 13).

4. The authors suspected that the S-type anion channels in the mutant might release more Cl⁻ from the guard cells into the medium. This hypothesis was tested with the non-specific animal anion channel pharmacological agent 9-AC.

Comment: to confirm their finding the authors by more direct evidence via e.g. proper EDXA analysis of K and Cl relative content.

If we could determine the relative contents of K and Cl in guard cells, it is very nice. However, if we determine the contents of K⁺ and Cl⁻ in guard cells by EDXA, we cannot see the exclusive release of Cl⁻ from guard cells because K⁺ is concomitantly released with efflux of Cl⁻. We will see the decrease of both Cl⁻ and K⁺, but we think that such data are not essential for this study. We believe that the patch clamp data in Fig. 3 support the release of Cl⁻.

5. The authors found apertures were smaller in the *cbc1 cbc2 slac1-4* triple mutant than in *slac1-4*. This result is probably due to the activation of SLAH3 in the mutant. Likewise, stomatal apertures were smaller in *cbc1 cbc2 slah3-3* than in *slah3-3* (Fig. 2d), probably due to activated SLAC1 in the triple mutant.

Comment: this postulate can be tested by guard cell patch clamping using Cl⁻ vs NO₃⁻ to distinguish between the contribution of SLAC1 and SLAH3.

Thank you for the comment. Although the contribution of SLAH3 to regulate stomatal aperture cannot be estimated correctly in this study (e.g. *flg22* signaling

reported by Deger et al., 2015), Negi et al., (2008) and Yamamoto (2016) reported that single SLAC1 disruption is enough to confer strong CO₂/bicarbonate-insensitive phenotype. We appreciate the reviewer's comment. However, the discrimination of contribution of SLAC1 and SLAH3 to stomatal aperture is out of our main focus because we tested only the effect of Cl⁻ on stomatal aperture and found the mutant phenotype presented in Fig. 2.

(References)

Negi et al., Nature (2008) 452: 483-486

Yamamoto et al., Plant Cell (2016) 28: 557-567

Guzel Deger et al., New Phytol. (2015) 208: 162-173

6. WT and *cbc1 cbc2* Guard Cell Protoplasts showed no large anion currents under control conditions, as has been reported (Ref.47,50). The activation of anion currents by bicarbonate was observed in GCPs from the *cbc1 cbc2* double mutant (Fig. 3a, c). However, in contrast to WT GCPs, the enhanced current of *cbc1 cbc2* GCPs was not suppressed by BL.

Comment 1: what about RL? Does RL inhibit calcium-induced S-type anion currents in GCPs? Mori et al., 2006 and Geiger et al., 2010 showed macroscopic anion currents in the presence of 2 μ M cytosolic free calcium. The authors need to show these calcium-induced S-type anion currents in the absence of RL as a control. Since several authors have spent time in the Schroeder lab and co-authored S-type anion channel studies, they should be aware of the fact that 2 μ M cytosolic free calcium is sufficient to activate S-type anion channels in GCPs and that these controls need to be shown in the absence of RL.

It is interesting to know the effects of RL on calcium-induced S-type anion channels. We, however, performed the experiments to isolate the BL effect on the anion current under the strong RL, and showed the current exhibited no significant difference between WT and the *cbc1 cbc2* mutant. We therefore think that no difference in the current was found between WT and the *cbc1 cbc2* mutant in response to RL.

According to your suggestion, we performed patch-clamp analysis with 2 μ M

[Ca²⁺]_{cyt} under dim white light where we routinely perform the experiments to see Ca²⁺ or ABA activation of S-type anion channels. We showed that the Ca²⁺-induced anion current exhibited no large difference between the *cbc1 cbc2* mutant and WT in the absence of RL. The data can be used for the controls. (Please see our response to next reviewer's comment). We note that pretreatment of GCPs with high extracellular CaCl₂ (40 mM) allows seeing Ca²⁺ activation of S-type anion channel currents in *Arabidopsis* (e.g. See Figure 1 of Mori et al., 2006).

Comment 2: The reader needs to know whether the calcium and ABA activation (without RL) of S-type channels (Schroeder and Hagiwara, 1989; Chen et al., 2010; Stange et al., 2010) in *cbc* mutants was WT like too?

As mentioned above, we added new results on the S-type anion channel that was activated by high extracellular Ca²⁺ mutant (Supplementary Fig. 6). The channel property was not affected in the double mutant. We did not try ABA, because the *cbc1 cbc2* mutant showed no ABA-related stomatal phenotype (Fig. 5b).

7. Phototropism directly phosphorylated CBC1 in vitro

Comment: Does BL lead to phosphorylation of SLAC1 and SLAH3 (Demir et al., 2013) in guard cells? Do phototropins directly phosphorylate the SLAC1 and SLAH3 channels?

It is interesting and important question. We would like to know it. We think that phototropins do not directly phosphorylate SLAC1 and SLAH3 channels in vivo. The phosphorylation of the channels may be mediated by CBCs or some other kinases, but such experiments are beyond our present work and should be done in future.

8. The authors claim that CBCs function downstream of HT1 and ultimately inhibit the S-type anion channels.

Comment: This hypothesis can be tested in the way J. Schroeder's lab did for the ABA-signaling cascade (Brandt et al., 2012).

This is a good suggestion. In this case, however, we should probably use the transmembrane domain of S-type anion channel as a substrate as has recently been reported (Yamamoto et al., 2016). We will do such type of experiments with collaboration in future.

9. The authors argue that it is possible that CBCs inhibit OST1 by phosphorylation, thereby suppressing S-type anion channels and/or protein kinases other than OST1 and causes CO₂-dependent stomatal closure by activation of SLAC1 through phosphorylation of the transmembrane region.
Comment: To answer this question the authors should include in their studies the OST1 mutant as well as CPK mutants.

Thank you for your suggestions. However, we just noted such possibilities in discussion to include the previous results and consider the future studies. The experiment suggested by reviewer seems to be beyond our focus. The experiments to clarify the relationships between CBCs and other components of OST1, RHC1, and CPK in guard cells should be done after the establishment of CBCs as the signaling components in the convergent point of blue light and CO₂.

10. Stomata in the *cbc1 cbc2* double mutant were clearly sensitive to ABA in epidermal peels, as has been reported for the *ht1* mutants and the authors claim CBCs do not play a role in ABA-induced stomatal closure.

SLAC1 and SLAH3 are activated in response to ABA and essential for ABA-dependent stomatal closure. CBCs are supposed to inhibit S-type channel opening, but the double mutant has a WT ABA response.

Comment: Shouldn't at least the *cbc1 ox* show an ABA insensitive phenotype? The reader would like to find these data?

We obtained the results, indicating that ABA sensitivity was not affected in the *cbc1 cbc2* double mutant. It is thus plausible that CBCs do not play a role in

ABA-induced stomatal closure, as has been reported for the *ht1* mutants. Since CBCs are not responsible for the ABA responses, we think that the suggested experiment raised by comment is not directly related to our present work.

11. S-type channels do not conduct malate. QUAC1 in Arabidopsis is required for malate induced stomatal closure. Malate is formed by PEP and CO₂.

Comment: Does the loss of *cbc1* and *2* affect QUAC1 activity?

It is possible that CBCs regulate the activity of QUAC1, and affect stomatal response. However, we obtained the phenotype that might be derived from the changes in Cl⁻ transport but not from malate transport in the assay of stomatal aperture (Fig. 2). Patch clamp experiments showed that such phenotype could be mainly accounted by the regulation of S-type channel activity (Fig. 3). We therefore think that to test the regulatory role of CBCs in QUAC1 is not always required for the present work and can be the future work.

12. SFig. 4(d) BL- and Fc-dependent H⁺-pumping from GCPs and (f) Fc-induced stomatal opening in epidermal peels of WT and *cbc1 cbc2*.

Comment: These data lack statistics, which should be provided.

We performed the statistic analysis for the responses, and the data are shown in Fig. 5d (former Fig. 4d) and Fig. 5f (former Fig. 4f).

13. Figure 2. The title of this figure is: "Enhancement of S-type anion channel activity in *cbc1 cbc2* mutants." Note, that title is misleading, as changes in anion channel activity are only shown in Fig. 3.

Thank you very much. We did not recognize it until you pointed it out. We changed the title as " S-type anion channels are involved in the reduced light-induced stomatal opening of *cbc* mutants".

14. Since the authors did not use heterologous expression system such as *Xenopus* oocytes, they should in the discussion mention that future reconstitution studies would be necessary to underpin their findings.

We briefly mentioned the future reconstitution studies using *Xenopus* oocyte as: To verify these findings, the reconstitution studies in *Xenopus* oocytes that express these components, including phot1 or HT1, CBCs, S-type channels, and OST1 are necessary.

Armstrong, F., Leung, J., Grabov, A., Brearley, J., Giraudat, J., and Blatt, M.R. (1995). Sensitivity to abscisic acid of guard-cell K⁺ channels is suppressed by ABI1-1, a mutant *Arabidopsis* gene encoding a putative protein phosphatase. *Proc. Natl. Acad. Sci. U. S. A.* 92, 9520-9524.

Brandt, B., Brodsky, D.E., Xue, S., Negi, J., Iba, K., Kangasjarvi, J., Ghassemian, M., Stephan, A.B., Hu, H.H., and Schroeder, J.I. (2012). Reconstitution of abscisic acid activation of SLAC1 anion channel by CPK6 and OST1 kinases and branched ABI1 PP2C phosphatase action *Proc. Natl. Acad. Sci. U. S. A.* 109, 10593-10598

Chen, Z.H., Hills, A., Lim, C.K., and Blatt, M.R. (2010). Dynamic regulation of guard cell anion channels by cytosolic free Ca²⁺ concentration and protein phosphorylation. *Plant J.* 61, 816-825.

Demir, F., Horntrich, C., Blachutzyk, J.O., Scherzer, S., Reinders, Y., Kierszniowska, S., Schulze, W.X., Harms, G.S., Hedrich, R., Geiger, D., and Kreuzer, I. (2013). *Arabidopsis* nanodomain-delimited ABA signaling pathway regulates the anion channel SLAH3. *Proc. Natl. Acad. Sci. U. S. A.* 110, 8296-8301.

Lebaudy, A., Vavasseur, A., Hosy, E., Dreyer, I., Leonhardt, N., Thibaud, J.B., Very, A.A., Simonneau, T., and Sentenac, H. (2008). Plant adaptation to fluctuating environment and biomass production are strongly dependent on guard cell potassium channels. *Proc. Natl. Acad. Sci. U. S. A.* 105, 5271-5276.

Schroeder, J.I., and Hagiwara, S. (1989). Cytosolic calcium regulates ion channels in the plasma-membrane of *Vicia faba* guard-cells. *Nature* 338, 427-430.

Schwartz, A., Wu, W.H., Tucker, E.B., and Assmann, S.M. (1994). Inhibition of inward K⁺ channels and stomatal response by abscisic acid - an intracellular locus of phytohormone action. *Proc. Natl. Acad. Sci. U. S. A.* 91, 4019-4023.

Stange, A., Hedrich, R., and Roelfsema, M.R.G. (2010). Ca²⁺-dependent activation of guard cell anion channels, triggered by hyperpolarization, is promoted by prolonged depolarization. *Plant J.* 62, 265-276.

Szyroki, A., Ivashikina, N., Dietrich, P., Roelfsema, M.R.G., Ache, P., Reintanz, B., Deeken, R., Godde, M., Felle, H., Steinmeyer, R., Palme, K., and Hedrich, R. (2001). KAT1 is not essential for stomatal opening. *Proc. Natl. Acad. Sci. U. S. A.* 98, 2917-2921.

Reviewer #3 (Remarks to the Author):

Blue light and CO₂ signals converge to regulate light-induced stomatal opening.

Review: Guard cells are endowed with sensor systems that allow them to fine-tune stomatal apertures in response to changes in light intensity, carbon dioxide concentration, temperature, and drought. This fine tuning serves to keep the optimal balance of photosynthesis with transpiration for a given set of environmental conditions and protect against excessive water loss under drought in high temperature conditions. It is well known that blue light induces stomatal opening by the activation of the phototropins and that high carbon dioxide induces stomatal closure. Conversely low carbon dioxide induces stomatal opening. Although there is a very large and sophisticated literature derived from electrophysiological studies on the channels and ion pumps involved, there remain serious gaps in knowledge of the signal-transduction pathways functioning with changes in carbon dioxide concentration and with phototropin activation.

Hiyama et al. characterize a pair of kinases, CBC1 and CBC2 that negatively regulate the two slow anion channels SLAC1 and SLAH1 in response either to low carbon dioxide or to blue light. The function of these channels is essential for stomatal closing. A *cbc1 cbc2* double mutant was severely impaired in stomatal

opening. In vitro, CBC1 was shown to be phosphorylated by an active phot1 preparation (both in-vitro pull-down assays and in vivo BiFC indicated a direct association of CBC1 with phot1. Curiously, the guard-cell pathway involving the phototropins, BLUS1, and a proton ATPase was unimpaired. A mix of electrophysiological, genetic, and physiological experiments with leaves/epidermal peels strongly implicated blue-light-activated CBC1 and CBC2 in negatively regulating SLAC1 and SLAH1 and consequently inhibiting stomatal closure.

HT1 is another kinase known to be a negative regulator of carbon dioxide-induced stomatal closure, the author used genetic experiments with *cbc1*, *cbc2*, and *ht1* mutants to show that they functioned on the same pathway to inhibit stomatal closing. They were then able to show both by in vitro pull-down assay and in vivo BiFC that these proteins interacted directly. Thus either light-activated phototropins, or low carbon dioxide-activated HT1 phosphorylated CBC1 and induce association with CBC2. The results are the first demonstration of components shared between these two pathways. I find this study a really solid advance in our knowledge of stomatal regulation by two different environmental parameters, establishing a common pathway, and recommend publication. I do have some minor comments however:

1. HT1 needs a little more explanation in the abstract.

We noted that HT1 as “HIGH LEAF TEMPERATURE 1” in the abstract.

2. P. 6, par. 1, l. ...H⁺-ATPases, respectively, reach...

Thank you. According to this comment, we improved this sentence as “phosphorylation levels of phototropin and the H⁺-ATPase, respectively, reach their maximum at these time points”.

3. P7. Par. 2. L. 2: ...but substantially to blue light... (delete and largely)

Thank you. According to your suggestion, we changed the sentences from

“and largely” to “but substantially to BL”.

4. P. 9, l. 2: “We hypothesize that the gene disruption....” L. 4: To test this hypothesis directly...” As written, text suggests author bias toward a favorable result.

According to your suggestion, we changed the sentence from “To show this phenomena directly” to “To test this hypothesis directly”.

5. In two places, “pull-downed” should be replaced with “pulled down”

We corrected them. We used “pulled down”. Thank you.

6. P. 13, par. 2, last line: use “relevance” here not “meaning”

According to your suggestion, we changed the word “meaning” to “relevance”.

7. Fig. 1 legend, epidermises, not epidermis. Epidermes. The images in 1f were not easy to see. Fig. 1g, why standard deviation? What differences are significant?

Thank you. We used “epidermes” as a plural form of epidermis instead of epidermises. The former Fig. 1f was changed to Fig. 1g. I agree that the images were not easy to see. However, we can see that the qualitative restoration of leaf temperature decrease in response to blue light was proportional to the expression levels of CBC1 and CBC2.

The former Fig.1g was changed to Fig.1h. We noted the statistical analyses and the meaning of the difference in the legends as. “Asterisks denote the significant differences in stomatal apertures between WT plants and mutants under darkness, RL, or RL+BL. * $P < 0.05$ by Student’s t test.”

8. Fig. 1 legend, last line: A sample of four is insufficient to calculate a meaningful standard error.

We repeated the experiments. We replaced “ $n=4$ ” by “ $n=8$ ”.

9. Figures 4d and 6a: These BiFC images merely establish that there is interaction in the periphery of guard-cell protoplasts, not that the interacting components are “in the plasma membrane.”

According to your suggestion, we removed the words “in the plasma membrane” and added in MCPs as “in the periphery of MCPs”. Sorry about to misled you that we used mesophyll cell protoplasts (MCPs) instead of GCPs in this experiment.

Reviewer #4 (Remarks to the Author):

The manuscript "Blue light and CO₂ signals converge to regulate light-induced stomatal opening ", submitted by Hiyama et al. describes two new kinases that link blue light and CO₂ induced stomatal opening. The results are original and are of high interest to the scientific community. The mechanism of activation of CBC1 and CBC2 and their regulation of S-type anion channels in response to both signaling pathways need however more work to support the claims presented by the authors.

We agree with this general comment. However, the present work focuses on the finding of new protein kinases that act as a convergent point between blue light and CO₂. It is of course important to address the activation mechanism of CBC1, CBC2 and their regulation of S-type anion channels, but the regulation of S-type channels by other protein kinases, including OST1 and CPK, remains unknown. To elucidate the activation mechanisms of CBCs and their regulation of S-type channels can be future works. Meanwhile, we did some experimental works according to the comments by this reviewer below.

In detail these questions should be addressed prior to resubmission:

Phosphorylation data

A potential mode of CBC regulation might be the phosphorylation of CBC1 or potentially CBC1/CBC2.

1) The authors do not mention how many biological replicates were performed for the phosphoproteomic study to quantify CBC1 phosphorylation.

We used 4 replicates of the preparation of guard cell protoplasts, and noted this in the legend of Fig. 1b.

2) No protein database, settings or statistical data are given for the quantitation of phosphorylation changes as well as for the peptide identification.

We added the name of database and other information in Materials and Methods. Quantitation of phosphorylation changes was done using Mass Navigator V1.3 as noted in Materials and Methods.

3) Several ions, including y11 and y14 are not marked in the graph in Figure1a.

Since these peaks are deaminated or dehydrated ones, we did not assign them. The peaks corresponded to y11 and y14 are not essential for the determination of peptide sequences and phosphorylation residues, and we did not mark them in Fig. 1a.

4) The authors could mention that the identified sites in CBC1 have been published before (see PhosPHAT 4.0).

We searched the PhosPhAt 4.0, and found that CBC1 was phosphorylated at the same sites. We wrote in the text as “We note here that the phosphopeptides from the same protein were reported in PhosPhAt 4.0 database.”

5) Was there only a doubly phosphorylated peptide of CBC1 identified or did the singly phosphorylated peptides not change in abundance?

We found both doubly and singly phosphorylated peptides. However, we repeatedly found the doubly phosphorylated peptides of CBC1. We could not determine the changes in abundance of the singly phosphorylated peptides.

6) Did the author quantify nonphosphorylated peptides of CBC1 to rule out changes in protein abundance?

We did not quantify the nonphosphorylated peptides of CBC1. However, we determined the abundance of CBC1 by immunoblot analysis using specific antibodies in response to blue light and showed no abundance change of CBC1 proteins (See Fig. 1b, Inset).

7) The raw data should be deposited in a common proteomics dataset, e.g. PRIDE, so that others, including the reviewers, can confirm the results.

I agree with your comment. However, we would like to keep our raw data for a while because we are now analyzing other targets. We will deposit the raw data at the time of the next publication.

8) In the figure caption of figure 1c, it says schematic structures indicate phosphorylation sites of CBC1 and CBC2. I do not find those in the actual figure.

Thank you. The figure caption presented was not good, and it misled you. We improved it in figure legend.

Thermal imaging

1) The main text claims that the cbc1 mutant is impaired in response to BL. This is very hard to judge based on Figure 1d. Thermal imaging should be quantified with multiple replicates and statistically analyzed to support those claims.

I agree with you. According to your comments, we presented the quantitative data as Figure 1e.

Stomata assay

1) The authors claim that BL dependent stomatal opening was slightly impaired in the single mutants of cbc1 and cbc2. Without a statistical analysis this cannot be judged from Figure1g.

Fig.1g was moved to new Fig. 1h. We performed the statistical analyses and presented as Fig. 1h.

I might also have misunderstood the statement in the introduction that RL induced stomatal opening is not apparent in epidermal peels of Arabidopsis. Because why otherwise would the authors measure RL induced stomatal opening in Figure1g?

You did not misunderstand. We explained this below.

There are two reasons to apply strong RL to the epidermis. One is to isolate BL-dependent stomatal opening from RL-induced (photosynthesis-dependent) one. The second is that strong RL is required to enhance BL-dependent stomatal opening in epidermis (and intact leaves). If we used only BL, no large opening is found in the epidermal peels (Shimazaki et al., 2007).

2) Stomatal conductance measurements in Figure1h need a statistical analysis. Normalization to the same starting conductance might also be a good idea to claim RL and BL effects.

Fig.1h is moved to Fig. 1i.

We repeated the same experiments and presented new data (Fig.1i). You can see that the error bars were shortened. On the basis of the new data, we

performed the statistical analyses with Supplementary Table 2. We then normalized them but presented the raw data because the raw data suggested that stomata closed tighter in the *cbc1 cbc2* double mutant than in WT plants under the dark (We noted this in the text according to minor comment by reviewer1, See Supplementary Table 1). The normalized data were presented in Supplementary Fig. 4.

3) Figure2a and Figure2b contain asterisks, indicating some kind of statistical test, which is not mentioned in the figure caption or main text. Please include a statistical analysis.

We added the details in legends.

4) As a general note, if statistical analyses of stomata assays are performed, keep in mind that stomatal opening and closing does not follow a normal distribution, care should be taken in selecting a statistical test.

We selected randomly the target stomata.

5) The authors claim that stomatal apertures were smaller in the *slah3cbc1ccb2* triple mutant than in *slah3*, probably due to activation of SLAC1. Based on the findings presented in the manuscript, SLAH3 has no phenotype as compared to wild-type plants, and the triple mutant merely reflects the phenotype of the *cbc1cbc2* double mutant.

It is true. We could not see the clear phenotype in *slah3* single mutant. We thus soften description of the role of SLAH3 in stomatal closure in the text.

6) 1mM CO₂ appears quite excessive for stomata assays. I know it was used before, but if I am not mistaken, 1 mM CO₂ calculates to 22.000 ppm (http://hiq.linde-gas.com/en/specialty_gases/gas_concentration_units_converter.html). Comments would be appreciated.

We used 1.2 mM bicarbonate salt (1 mM free CO₂, calculated with Henderson–Hasselbalch equation) to activate S-type anion channels for patch clamp analysis, but NOT for stomatal bioassay experiments. Please note that Hu et al., (2010) used 1.2 mM or 2.4 mM bicarbonate (1 mM or 2 mM free CO₂ respectively) in extracellular solution to activate S-type anion channel. In addition, Xue et al., (2011) and Tian et al., (2015) used 13.5mM HCO₃⁻ (2mM free CO₂) in "intracellular solution. We also tried low bicarbonate concentration (0.3 mM bicarbonate salt) but no significant S-type current was observed in WT (data not shown). As suggested by Hu et al., (2010), one possible reason for this is that cytosolic diffusible small molecules or proteins may be required for full sensitivity of bicarbonate/CO₂ because whole-cell patch clamp includes dialysis of the cytoplasm (Hamill et al., 1981).

(References)

Hu et al., (2010) Nat Cell Biol. 12: 87-93

Xue et al., (2011) EMBO J 30:1645-1658

Tian et al., (2015) Nat Commun 6: 6057

Hamill et al., (1981) Pflugers Arch 391: 85-100

Electrophysiology

1) No statistics on the electrophysiological assays.

Thank you for pointing out this. As mentioned in response to reviewer 1, now we show statistical analysis results for the data in the figures. Different letter labels indicate significant differences ($p < 0.05$, Tukey-Kramer test).

Phosphorylation and kinase assays

Besides the phosphoproteomic data presentation, that lays the foundation of a putative phospho-regulation of CBC proteins, the kinase assays are the biggest problems of the manuscript. The authors do not show that phototropin or HT1 phosphorylate CBC proteins at the site they identified in their screen. The

presented experiments do not sufficiently support a regulation by phototropin or HT1. Phosphomimic mutants, in vivo phosphorylation analyses in respective phototropin or *ht1* mutant lines might help to further substantiate such claim.

(A) We showed the phosphorylation sites of CBC1 by phototropin1 in vivo in Fig. 1c. We further showed that CBC1 was phosphorylated by phot1 in vitro, but it is unclear whether the phosphorylation sites in vitro are the same as those in vivo. To test this, we replaced two in vivo phosphorylated Sers (S43, S45) by Alas in CBC1 and utilized this protein as substrate for P1C. We found no phosphorylation of the CBC1 by phototropin1 (Fig. 4f), suggesting that the sites in vitro are the same as those in vivo or contain one of them. We added new data in the text.

(B) We tried to determine the phosphorylation sites of CBCs by HT1 in vivo. Since there is no clue to activate HT1 specifically, we tried to activate HT1 kinase by changing the concentrations of CO₂ (low and high CO₂). However, we could not see the change in phosphorylation levels of CBC1 and HT1 in vivo by Phos-tag PAGE. We further determined the phosphorylation levels of CBC1 in both *ht1* mutant and WT, but we could not see the difference in phosphorylation levels of CBC1 between them. Since substrate and activation signal for HT1 have not been determined so far, we think that to identify in vivo phosphorylation sites of CBCs by HT1 can be the future work.

By contrast, we showed the phosphorylation of CBC1 by HT1 in vitro (Fig. 6e). When phot1-mediated phosphorylation sites of two Sers (S43, S45) in vivo were replaced by Alas, the CBC1 was further phosphorylated by HT1 in vitro. The results suggest that there may be additional or different phosphorylation sites of CBC1 by HT1 compared with those by phot1. We presented new data in Fig. 6f.

(C) To use the phosphomimic mutants of CBCs in phototropin- or HT1-pathway, we have to determine all the phosphorylation sites of CBCs in vivo by phot1 and HT1. It is not easy and takes very long time to prepare the sufficient amount of guard cell proteins for phosphoproteomic analysis. Furthermore, if we succeeded in the determination of all phosphorylation sites and the elucidation

of activation mechanisms for CBCs, phosphomimic mutants do not always work and depend on the protein kinase species. For this reason, we did not perform the phosphomimic experiments in the mutant lines.

1) The Phostag shift in Figure 4a requires an additional phosphatase control to confirm this shift is due to phosphorylation.

This is essentially the same question on Figure 4b, and we reply to both comments here. (A) We first describe the reason that the Phos-tag shift is likely due to the phosphorylation of CBC1. (B) Second, we performed three lines of experiments to address for this question using phosphatase, but unfortunately we could not obtain the positive results. We wrote these experiments. (C) Third, we obtained the positive results using the transgenic plants. We noted this result and included it in the text.

(A) Phos-tag acts as the phosphate chelator causing the reduction of mobility in proteins and is able to detect the protein phosphorylation. It is possible that some other BL-induced modification of CBC1 also bring about the reduction of the mobility. However, we did not find the mobility shift of CBC1 in response to BL when CBC1 proteins were subjected to SDS-PAGE (See Fig. 1b, Inset), suggesting that the mobility shift is specific to Phos-tag. In addition to this, we provide evidence that phot1 kinase phosphorylates CBC1 in vivo and in vitro and that phot1 interacts with CBC1 in vivo and in vitro.

(B) We investigated the effect of phosphatases on the mobility shift in three lines of experiments (a) After the treatment of GCPs by light, we obtained CBC proteins by the addition of TCA (trichloroacetic acid) to GCPs. We applied bacterial alkaline phosphatase (BAP) to the precipitated proteins that had been resuspended using voltex mixer, but we could not obtain clear results probably due to that BAP could not reached to the target CBC1 proteins. (b) We expressed CBC1-GFP in guard cells, and after the illumination of GCPs with BL CBC1-GFP was immunoprecipitated by GFP (GFP trap), but mobility shift of the immunoprecipitated CBC1-GFP was not clearly found because the protein

profiles of CBC1-GFP were diffused on Phos-tag SDS-PAGE. We applied BAP or lambda phosphatase to the precipitate, but we could not see the mobility change. (c) After the BL treatment of GCPs that expressed CBC1-GFP, we disrupted GCPs with detergent in the presence of phosphatase inhibitors (CalyclinA or lambda phosphatase) and the cell lysate was centrifuged to obtain the supernatant. Then, we obtained CBC1 proteins by TCA treatment from the supernatant, and subjected to Phos-tag PAGE. However, we could not see the mobility shift of CBC1 in response to BL.

All these experiments were done using GCPs, but most of the experiments to show the phosphatase effect was obtained in cell extracts or isolated proteins. We think that to see the phosphatase effect in the cells may be difficult. For example, recent work that shows the phosphorylation of SLAC1 by Phos-tag PAGE did not show the effect of phosphatase (Yamamoto et al., 2016, Plant Cell).

(C) Phosphoproteomic analysis revealed that phot1 phosphorylated Ser43 and Ser45 in vivo. phot1 phosphorylated CBC1 in both in vitro and in vivo, but it is unclear whether the phosphorylation sites in vitro are the same as those in vivo. To test this, we replaced two in vivo phosphorylated Sers (S43, S45) (Fig. 1c) by Alas in CBC1 (D271N) and utilized this protein as substrate for P1C in vitro. We found no phosphorylation of the mutant CBC1 (Fig. 4f), suggesting that the sites in vitro are the same or one of them.

We thus expect that the mutant CBC1 (S43A S45A) will neither show the phosphorylation nor mobility shift in vivo in response to blue light. In accord with this, when the *CBC1 (S43A S45A)-GFP* was expressed in the *cbc1 cbc2* double mutant (Fig. 4g), no mobility shift of the mutant CBC1 (S43A S45A)-GFP was found although WT CBC1-GFP exhibited the typical shift in response to blue light (Fig. 4g). The result confirms that the CBC1 (S43A S45A) cannot be phosphorylated in guard cells and that the mobility shift is due to phosphorylation (Fig. 4a,b).

2) CBC2 does show multiple bands in Figure4a. These might be different phosphorylation forms. The second one from the bottom might even increase in response to BL. Could the authors comment.

This is interesting question. We performed the experiments three times, but we could not reproduce the phosphorylation of second band from the bottom in WT, but we are not sure on the band in the *cbc1* mutant. We presented two data as Supplementary Fig. 7.

3) Figure4b is not convincing and minimum requires a phosphatase treatment.

We replied to this comment as above.

4) The kinase assay in Figure1e contains more enzyme (P1C) than substrate (CBC). Under these conditions nearly every Ser/Thr kinase can phosphorylate CBC.

Figure 1e may be former Fig. 4e. I cannot understand why reviewer thought the presence of excess enzyme (P1C). On the basis of CBB staining in the lower panel of Fig. 4e, the amount of P1C is lower than that of CBCs. We previously used the same method on BLUS1 protein (Takemiya et al., 2015, Plant Cell Physiol).

5) In general, since the authors are able to quantify phosphorylation changes in planta. Why don't they do this for *ht1* and *phot* mutants.

We used *phot1 phot2* double mutant as shown in Fig.1. It is not easy to perform the quantitative phosphoproteome analyses on the responses specific to guard cells because provision of the sufficient amount of GCPs is usually difficult. So we attempted to compare the phosphorylation levels of CBCs in *ht1* mutant and WT plants through Phostag PAGE, and stimulated guard cells by changing CO₂ concentration stimuli. However, we could not find the mobility shift. Since environmental stimuli for the activation of HT1 are not identified so far and the

substrate(s) for HT1 remain unclear, it will take some more time to determine the phosphorylation sites of CBCs by HT1.

Reviewers' comments:

Reviewer #1 (Remarks to the Author):

In conclusion this is an interesting and important manuscript that reports relevant new findings that will be of significant interest to the community. The authors have performed the experiments suggested by this reviewer to satisfaction. The authors have provided detailed and sufficient responses to this reviewer's comments (reviewer 1) and to the comments of the other reviewers, based on this review. This reviewer agrees with the responses provided by the authors and agrees that some analyses will need to be done in the future, and may take some years to complete. The findings reported in this manuscript will be of great interest to the community, as the manuscript reports the discovery of important new genes with very interesting well characterized and relevant blue light and CO₂ signaling mutant phenotypes and depth in analyses is included. Publication in Nature Communications is strongly supported.

Some minor comments that the authors could consider, but that are not required for this manuscript, are listed in the following:

As a negative control for BiFC experiments, testing of a protein pair that is expressed in the same cellular compartment but is known not to interact would be best. As weak BiFC interaction fluorescence can occur in such negative control cases, quantifying of BiFC fluorescence intensities and showing averages is recommended. Since the authors have performed pull-down assays for the concluded interacting proteins, these experiments may not be essential in this paper though.

The authors respond to a comment by adding the sentence: "To verify these findings, the reconstitution studies in *Xenopus* oocytes that express these components ... are necessary." The term "verify" appears a bit strong here. Although oocytes provide a powerful reconstitution tool to generate and provide a first test of a hypothesis, in this reviewer's view they may not always "verify" an in planta model.

Non-anonymous review signed by Julian Schroeder & Jingbo Zhang

Reviewer #2 (Remarks to the Author):

The identification and characterization of CBC proteins in guard cells is an important contribution to the field. I thus recommended the paper to be accepted for publication in Nature Communication, pending a number of points of concern that needed attention. Below I will discuss how the authors have dealt with my comments and also list a number of remarks for improvement of the manuscript.

Previous comments.

1. In the previous version of the manuscript, data were lacking to support the statement that CBC1 and CBC2 are expressed in guard cells. The authors now show PCR data that confirm the presence of CBC1 and 2 transcripts in guard cells and they transformed the *cbc1/2* double mutant with GFP tagged CBCs constructs. These data are shown in Suppl. Fig. 3. Unfortunately, it is neither mentioned in the text, not in the legend, which promoters were used to express CBC1 and 2. This information should be added and a negative control of the fluorescence of guard cells has to be included in Suppl. Fig. 3b.

2. The results obtained with the gas-exchange technique varied to some extent from those of stomatal apertures measurements in epidermal strips. The authors explain in their reply, as well as in the text of the manuscript that stomatal responses in intact leaves often differ from those in epidermal

strips and point to a potential role of the mesophyll in responses of stomata to PAR. This discrepancy between IRGA on intact leaves and isolated epidermal peels has to be clearly mentioned in the text.

3. A 50 % reduction of the inward K⁺ channel activity was found for the *cbc1/2* double mutant, but the authors wrote in the first version of the manuscript that this will not strongly effect stomatal movements. They now explain in more detail in the discussion, why the reduction in K⁺ channel activity is unlikely to affect stomatal movements.

4. The authors found that stomata of the *cbc1/2* mutant tend to close more rapidly in blue light as those of wild type. They could abolish this difference by including the broad range anion channel inhibitor 9-AC in their assays. I proposed to use the EDXA technique to confirm that guard cells of *cbc1/2* mutants are releasing Cl⁻ at a faster rate as wild type. The authors explain that it is difficult to quantify Cl⁻ in guard cells with the EDXA technique, as guard cells will lose Cl⁻ concomitantly with K⁺. I do not fully understand this argument. However, the authors are correct that they directly show that CBC1 and 2 are essential for blue light-dependent inhibition of S-type anion channels in Fig. 3. The data of Fig. 3 indeed support an enhanced release of anion from *cbc1/2* guard cell in blue light.

5. It is shown that CBC1 is important for the blue light-dependent inhibition of S-type anion channels in guard cells, but it remains unclear if blue light targets SLAC1, SLAH3, or both S-type anion channels. In their rebuttal, the authors explain that examination of the relative roles of SLAC1 and SLAH3 was not within the scope of their study. I understand that this issue is not yet covered in the manuscript, but in this case the authors also should step back from discussing the relative roles of SLAC1 and SLAH3 for stomatal closure, as they do in the results (lines 235-236). Please note that there is no general agreement on this issue. Whereas the data of Vahisalu et al. (2008) and Negi et al. (2008) indicate a major role for SLAC1, Guzel Deger et al., (2015) only found that ABA-induced stomatal closure was only impaired in the *slac1/slah3* double mutant, but not in single mutants. Possibly, the relative contribution of SLAC1 and SLAH3 to anion release depends on the conditions that are used to grow Arabidopsis plants.

6. The patch clamp studies were conducted with 2μM free Ca²⁺ in the pipette, which should be sufficient to activate CPKs that in turn stimulate S-type anion channels. Nevertheless, only a weak activity of S-type channels was observed in the absence of bicarbonate. I wondered if red light inhibited the activity of S-type anion channels under control conditions. The authors do not directly answer this question, but instead they show that high extracellular Ca²⁺ levels can activate S-type anion channels, just as bicarbonate.

In addition, the reader would like to know if ABA still can activate S-type anion channels in the *cbc1/2* mutant. The latter issue was not answered directly, but the authors do show that ABA can still induce stomatal closure in the *cbc1/2* mutant (Fig. 5b).

7 and 8. Do phototropins and/or CBC proteins phosphorylate and inhibit S-type anion channels in guard cells? The authors explain that it is unlikely that phototropins directly phosphorylate SLAC1 or SLAH3, but would like to test if CBC proteins regulate these anion channels in future experiments.

9. In the discussion, the authors suggested that CBC1 and 2 regulate OST1. This could be tested by crossing the *cbc1/2* and *ost1* mutants. The authors explain that this experiment is beyond the scope of the current study. Given that the authors neither want to test whether CBCs phosphorylate SLAC1 and/or SLAH3I and resist testing OST1 phosphorylation, there is not much of the CBC signaling pathway understood. Issues remaining just speculative from referees point of view does not deserve discussion.

10. Since CBC1 and 2 are supposed to inhibit the activity of S-type anion channels, over-expression of their genes should inhibit ABA-induced stomatal closure. The authors explain that this experiment is not directly related to the topic of their study.

11. In addition to S-type anion channels, guard cells also possess R-type anion channels and I wondered if these channels are regulated by CBC1 and 2. This issue is regarded as a good topic for future work.

12. The data in Suppl. Fig. 5D (former Suppl. Fig. 4d) were presented without any statistics. In the current Suppl. Figure 5D these data are added.

13. The title of Fig. 2 was misleading. It has now been changed.

14. The *Xenopus* oocyte expression system is ideal to test the role of CBC1 and 2 in blue light signaling, with reconstitution studies. This is now mentioned in the discussion of the manuscript.

Remarks:

2. Abstract, lines 20-21. As the CBC1/2 proteins are first identified in the current study it would be good to write: "Here, we identify and characterize CBC1/2 ...".

3. Abstract, line 24. Instead of "... by negatively regulating.." one can also write "... by inhibition of ...".

4. Abstract, line 29. As PAR to some extent may act through changes in the intercellular CO₂ level, it may be better to write: "...signals from BL and low CO₂."

5. Introduction, line 44. As the Polypodiopsida are and a clade of the ferns, please write. "... expect for fern species belonging to the Polypodiosida".

6. Introduction, line 82-83. Two recent studies of Horak and et al. (2016) and Jakobson et al. (2016) show that regulation of HT1, by MPK12 and 4, is essential for CO₂-signaling in guard cells. Please include this information in the manuscript.

7. Results, line 117-119. This sentence is very hard to read. Please improve

8. Results, line 145-146. CBC1 and 2 are closely related to HT1 (At1G62400). In Suppl. Fig. 2 HT1 is placed just 3 positions above CBC1 in the phylogenetic tree. Please mention this and discuss the evolutionary origin of the three genes in the discussion.

9. Results, line 154. What does the "s" stand for in "CBC1-sGFP" and "CBC2-sGFP"? In which respect are the data in Fig. 1F, lane 4 different from those in Fig. 1G?

10. Results lines 159-167. Please rearrange the text, so that the phenotypes of single mutants are discussed first and that of the double mutant thereafter.

11. Results line 205-206. Instead of writing "stomata further opened in WT plants and did not close in the double mutant.", it would be more logical to write: "stomata opened in WT, but the aperture remained constant in cbc1/2."

12. Results, line 228. It is stated that the results indicate that SLAC1 and SLAH3 genes are epistatic to

CBC1 and CBC2. It is therefore not logic to write that SLAC1 and SLAH3 likely function downstream of CBC1 and CBC2. Instead one might write: "... CBC 2 genes; this suggests that SLAC1 and SLAH3 are regulated by CBC1 and CBC2."

13. Results line 233-235. As explained above, in "previous comments number 5", there is no common sense about the respective roles of SLAC1 and SLAH3 in stomatal closure. In Fig. 2C, loss of cbc1 and 2 decreases the stomatal aperture, in the slac1-4 loss-of-function mutant (see 4 bars on the right). This indicates that CBC1 and 2 inhibit SLAH3, which does not fit a marginal role for SLAH3 in stomatal closure as stated in the text.

14. Results, line 265. The inset of Fig. 1B is not mentioned in the legend of this figure and no experimental details about these results are given.

Reviewer #5 (Remarks to the Author):

The manuscript "Blue light and CO₂ signals converge to regulate light-induced stomatal opening" by Hiyama et al., reports proteomic identification of a MAPKKK that is phosphorylated by phototropins in response to blue light. The authors go on to nicely demonstrate how CBC1(and2) regulate stomatal aperture.

Using a combination of in vivo (mass spectrometry in phot1 phot2 mutant; 1b) and in vitro kinase assays (fig 4) the authors convincingly demonstrate the phototropins phosphorylate CBC1 on S43/S45. Additionally, in vitro assays show that HT1 phosphorylates CBC1 and CBC2 though the sites of phosphorylation remain elusive, which I agree is beyond the scope of this manuscript. While the phosphorylation data is solid I do have several comments:

- Details for proteomics methods are insufficient. What is the FDR used for database search, # missed cleavages allowed, fixed modifications, mass tolerance ect. Additionally, should provide a supplemental file of all peptides identified, their MASCOT score, precursor charge and m/z, peptide identification score, which is standard for proteomic reports.
- Raw proteomics data needs to be deposited in a repository (MASSIVE, Proteome Exchange ect).
- No description of CBC1 or CBC2 antibodies.
- Panel 6F is not described in the figure legend.

Response to Reviewers' comments:

Reviewer #1 (Remarks to the Author):

In conclusion this is an interesting and important manuscript that reports relevant new findings that will be of significant interest to the community. The authors have performed the experiments suggested by this reviewer to satisfaction. The authors have provided detailed and sufficient responses to this reviewer's comments (reviewer 1) and to the comments of the other reviewers, based on this review. This reviewer agrees with the responses provided by the authors and agrees that some analyses will need to be done in the future, and may take some years to complete. The findings reported in this manuscript will be of great interest to the community, as the manuscript reports the discovery of important new genes with very interesting well characterized and relevant blue light and CO₂ signaling mutant phenotypes and depth in analyses is included. Publication in Nature Communications is strongly supported.

Some minor comments that the authors could consider, but that are not required for this manuscript, are listed in the following:

As a negative control for BiFC experiments, testing of a protein pair that is expressed in the same cellular compartment but is known not to interact would be best. As weak BiFC interaction fluorescence can occur in such negative control cases, quantifying of BiFC fluorescence intensities and showing averages is recommended. Since the authors have performed pull-down assays for the concluded interacting proteins, these experiments may not be essential in this paper though.

The authors respond to a comment by adding the sentence: "To verify these findings, the reconstitution studies in *Xenopus* oocytes that express these components ... are necessary." The term "verify" appears a bit strong here. Although oocytes provide a powerful reconstitution tool to generate and provide

a first test of a hypothesis, in this reviewer's view they may not always "verify" an in planta model.

According to your adequate suggestion, we softened the expression as: "The reconstitution studies in *Xenopus* oocytes that express these components, including phot1 or HT1, CBCs, S-type channels, and OST1 will confirm these findings in planta."

Non-anonymous review signed by Julian Schroeder & Jingbo Zhang

Reviewer #2 (Remarks to the Author):

The identification and characterization of CBC proteins in guard cells is an important contribution to the field. I thus recommended the paper to be accepted for publication in Nature Communication, pending a number of points of concern that needed attention. Below I will discuss how the authors have dealt with my comments and also list a number of remarks for improvement of the manuscript.

Previous comments.

1. In the previous version of the manuscript, data were lacking to support the statement that CBC1 and CBC2 are expressed in guard cells. The authors now show PCR data that confirm the presence of CBC1 and 2 transcripts in guard cells and they transformed the *cbc1/2* double mutant with GFP tagged CBCs constructs. These data are shown in Suppl. Fig. 3. Unfortunately, it is neither mentioned in the text, nor in the legend, which promoters were used to express CBC1 and 2. This information should be added and negative control of the fluorescence of guard cells has to be included in Suppl. Fig. 3b.

We mentioned this in the text as: When *CBC1-GFP* or *CBC2-GFP* was expressed in the *cbc1 cbc2* double mutant using each own promoter, GFP fluorescence was found in the cytosol of guard cells (Supplementary Fig. 3b). We also noted this in the figure legend of Supplementary Fig. 3b.

We also added a datum of a negative control for the GFP fluorescence in guard cells. We used guard cells of WT as a negative control, although ideal control should be guard cells of *cbc1 cbc2* double mutant. I think that WT can be the negative control for GFP fluorescence. The fluorescence is not due to autofluorescence as shown in Supplementary Fig. 3b.

2. The results obtained with the gas-exchange technique varied to some extent from those of stomatal apertures measurements in epidermal strips. The authors explain in their reply, as well as in the text of the manuscript that stomatal responses in intact leaves often differ from those in epidermal strips and point to a potential role of the mesophyll in responses of stomata to PAR. This discrepancy between IRGA on intact leaves and isolated epidermal peels has to be clearly mentioned in the text.

Thank you for your suggestion. We realized that we should note this important fact. We noted as: We note here that stomata exhibited the different responses between intact leaves and epidermis in *Arabidopsis*: stomata open in the former but scarcely open in the latter in response to RL (Fig. 1h, i). Such difference is partly due to the situation in which guard cells are placed. In leaves mesophyll tissues might provide guard cells with unidentified substances that stimulate stomatal opening but in epidermal peels such tissues are absent.

3. A 50 % reduction of the inward K⁺ channel activity was found for the *cbc1/2* double mutant, but the authors wrote in the first version of the manuscript that this will not strongly effect stomatal movements. They now explain in more detail in the discussion, why the reduction in K⁺ channel activity is unlikely to affect stomatal movements.

4. The authors found that stomata of the *cbc1/2* mutant tend to close more rapidly in blue light as those of wild type. They could abolish this difference by including the broad range anion channel inhibitor 9-AC in their assays. I proposed to use the EDXA technique to confirm that guard cells of *cbc1/2* mutants are releasing Cl⁻ at a faster rate as wild type. The authors explain that it

is difficult to quantify Cl⁻ in guard cells with the EDXA technique, as guard cells will lose Cl⁻ concomitantly with K⁺. I do not fully understand this argument. However, the authors are correct that they directly show that CBC1 and 2 are essential for blue light-dependent inhibition of S-type anion channels in Fig. 3. The data of Fig. 3 indeed support an enhanced release of anion from *cbc1/2* guard cell in blue light.

I misunderstood the word “relative content” in your original comment. Sorry about that. We thought it as the ratio of K⁺ to Cl⁻ in guard cells. If we can determine the content of Cl⁻ in guard cells by the EDXA technique, direct evidence will be obtained. However, I appreciate very much that you noticed the data of Fig. 3 could be an indirect evidence for the enhanced release of anion in *cbc1 cbc2* mutant.

5. It is shown that CBC1 is important for the blue light-dependent inhibition of S-type anion channels in guard cells, but it remains unclear if blue light targets SLAC1, SLAH3, or both S-type anion channels. In their rebuttal, the authors explain that examination of the relative roles of SLAC1 and SLAH3 was not within the scope of their study. I understand that this issue is not yet covered in the manuscript, but in this case the authors also should step back from discussing the relative roles of SLAC1 and SLAH3 for stomatal closure, as they do in the results (lines 235-236). Please note that there is no general agreement on this issue. Whereas the data of Vahisalu et al. (2008) and Negi et al. (2008) indicate a major role for SLAC1, Guzel Deger et al., (2015) only found that ABA-induced stomatal closure was only impaired in the *slac1/slah3* double mutant, but not in single mutants. Possibly, the relative contribution of SLAC1 and SLAH3 to anion release depends on the conditions that are used to grow *Arabidopsis* plants.

According to your comment, we eliminated the words “marginal” and replaced those of “not large in our plant materials”. We also noted that “the possibility that SLAC1 and SLAH3 redundantly function in stomatal closure.”

6. The patch clamp studies were conducted with $2\mu\text{M}$ free Ca^{2+} in the pipette, which should be sufficient to activate CPKs that in turn stimulate S-type anion channels. Nevertheless, only a weak activity of S-type channels was observed in the absence of bicarbonate. I wondered if red light inhibited the activity of S-type anion channels under control conditions. The authors do not directly answer this question, but instead they show that high extracellular Ca^{2+} levels can activate S-type anion channels, just as bicarbonate. In addition, the reader would like to know if ABA still can activate S-type anion channels in the *cbc1/2* mutant. The latter issue was not answered directly, but the authors do show that ABA can still induce stomatal closure in the *cbc1/2* mutant (Fig. 5b).

We did not investigate the effect of RL on the S-type channel activity in guard cells under our conditions. Since RL induces stomatal opening in intact leaves and the opening is impaired in the *cbc1 cbc2* double mutant, it is very important question. However, we believe that this question can be a separate subject from the present work, and should be investigated in more detail.

Although we did not determine whether ABA could activate the S-type anion channels, it is likely that ABA activates the channels because ABA caused stomatal closure in both *cbc1 cbc2* mutant and WT. Determination of S-type anion channels in response to ABA in the mutant can be the next work as the reviewer also recognize it.

7 and 8. Do phototropins and/or CBC proteins phosphorylate and inhibit S-type anion channels in guard cells? The authors explain that it is unlikely that phototropins directly phosphorylate SLAC1 or SLAH3, but would like to test if CBC proteins regulate these anion channels in future experiments.

We are also interested in this work as future experiments.

9. In the discussion, the authors suggested that CBC1 and 2 regulate OST1. This could be tested by crossing the *cbc1/2* and *ost1* mutants. The authors explain that this experiment is beyond the scope of the current study. Given that

the authors neither want to test whether CBCs phosphorylate SLAC1 and/or SLAH3 and resist testing OST1 phosphorylation, there is not much of the CBC signaling pathway understood. Issues remaining just speculative from referees point of view does not deserve discussion.

We agree with your comment. We deleted the sentence of “It is also possible that CBCs inhibit OST1 by phosphorylation, thereby suppressing S-type anion channels (Fig. 7).”

10. Since CBC1 and 2 are supposed to inhibit the activity of S-type anion channels, over-expression of their genes should inhibit ABA-induced stomatal closure. The authors explain that this experiment is not directly related to the topic of their study.

Since ABA-induced stomatal closure is not impaired in the *cbc1 cbc2* double mutant (Fig. 5b), we think that CBCs might not be responsible for the ABA-induced stomatal closure. Therefore, we believe that overexpression of CBC genes will not affect ABA-induced stomatal closure. However, this can be an interesting experiment because the result will provide clue to the interaction between the CO₂ and ABA signaling.

11. In addition to S-type anion channels, guard cells also possess R-type anion channels and I wondered if these channels are regulated by CBC1 and 2. This issue is regarded as a good topic for future work.

We agree with your comment.

12. The data in Suppl. Fig. 5D (former Suppl. Fig. 4d) were presented without any statistics. In the current Suppl. Figure 5D these data are added.

OK

13. The title of Fig. 2 was misleading. It has now been changed.

Thank you for your adequate suggestion.

14. The *Xenopus* oocyte expression system is ideal to test the role of CBC1 and 2 in blue light signaling, with reconstitution studies. This is now mentioned in the discussion of the manuscript.

Remarks:

2. Abstract, lines 20-21. As the CBC1/2 proteins are first identified in the current study it would be good to write: "Here, we identify and characterize CBC1/2 ...".

Thank you. We inserted the word "identify and" in the sentence according to your suggestion.

3. Abstract, line 24. Instead of "... by negatively regulating.." one can also write "... by inhibition of ...".

Thank you. We wrote as "stomatal opening by inhibition of S-type anion channels". It is clear and easy to understand.

4. Abstract, line 29. As PAR to some extent may act through changes in the intercellular CO₂ level, it may be better to write: "...signals from BL and low CO₂."

According to your suggestion, we replaced PAR by BL.

5. Introduction, line 44. As the Polypodiopsida are and a clade of the ferns, please write. "... expect for fern species belonging to the Polypodiopsida".

Thank you. We changed the expression as "the fern species belonging to the Polypodiopsida".

6. Introduction, line 82-83. Two recent studies of Horak and et al. (2016) and Jakobson et al. (2016) show that regulation of HT1, by MPK12 and 4, is essential for CO₂-signaling in guard cells. Please include this information in the manuscript.

Thank you for your comment. We included the information in the text as:
Recently, two mitogen-activated protein kinases (MPKs) MPK4 and MPK12 are shown to inhibit HT1 activity, thereby, stimulating stomatal closure in response to CO₂ (Horak et al., 2016; Jakobson et al., 2016).

7. Results, line 117-119. This sentence is very hard to read. Please improve.

We improved the sentence and described it in more detail in the text as: The method facilitates the identification of components that function redundantly because the mutant screening with a single mutation does not always provide clear phenotype.----

8. Results, line 145-146. CBC1 and 2 are closely related to HT1 (At1G62400). In Suppl. Fig. 2 HT1 is placed just 3 positions above CBC1 in the phylogenetic tree. Please mention this and discuss the evolutionary origin of the three genes in the discussion.

Thank you very much. It is very important suggestion and we did not realize this until you pointed it out. We mentioned this in the discussion as “We also noticed here that HT1 (AT1g62400) and CBCs belong to the Subgroups C5 and C7, respectively, in this phylogenetic tree of the MAPKKK multigene family (Supplementary Fig. 2). These three genes are likely derived from the same ancestral gene. Furthermore, CBC1 and CBC2 recently evolved by duplication.” We added acknowledgement for this discussion to Professor H. Tachida in the lab of evolutionary genetics.

9. Results, line 154. What does the “s” stand for in “CBC1-sGFP” and “CBC2-sGFP”? In which respect are the data in Fig. 1F, lane 4 different from those in Fig. 1G?

Thank you for your comment. It is our mistake. We used “sGFP” throughout this work. “sGFP” means “syntheticGFP” which is adapted for usage of eukaryote and is usually used for plant materials. But, we forgot to delete “s” from sGFP. So we just deleted these “s”.

10. Results lines 159-167. Please rearrange the text, so that the phenotypes of single mutants are discussed first and that of the double mutant thereafter.

According to your suggestion, we rearranged the sentences in the text as: BL-dependent stomatal opening was not affected in the single mutants of *cbc1* and slightly impaired in the *cbc2*, but severely in the *cbc1 cbc2* double mutant.

11. Results line 205-206. Instead of writing “stomata further opened in WT plants and did not close in the double mutant.”, it would be more logical to write: “stomata opened in WT, but the aperture remained constant in *cbc1/2*.”.

We rewrote it according to your suggestion.

12. Results, line 228. It is stated that the results indicate that SLAC1 and SLAH3 genes are epistatic to CBC1 and CBC2. It is therefore not logic to write that SLAC1 and SLAH3 likely function downstream of CBC1 and CBC2. Instead one might write: “... CBC2 genes; this suggests that SLAC1 and SLAH3 are regulated by CBC1 and CBC2.”

Thank you. According to your suggestion, we rewrote it as noted above.

13. Results line 233-235. As explained above, in “previous comments number 5”, there is no common sense about the respective roles of SLAC1 and SLAH3 in stomatal closure. In Fig. 2C, loss of *cbc1* and 2 decreases the stomatal aperture,

in the *slac1-4* loss-of-function mutant (see 4 bars on the right). This indicates that CBC1 and 2 inhibit SLAH3, which does not fit a marginal role for SLAH3 in stomatal closure as stated in the text.

We improved our expression from “marginal” to “not large in our plant materials” We also noted that “the possibility that SLAC1 and SLAH3 redundantly function in stomatal closure”.

14. Results, line 265. The inset of Fig. 1B is not mentioned in the legend of this figure and no experimental details about these results are given.

Thank you. We mentioned this in the bottom of this paragraph in a previous manuscript, but it was difficult to find out. We moved the similar sentences at the new lines of 132-133. We also noted this in Fig.1b legend.

Reviewer #5 (Remarks to the Author):

The manuscript "Blue light and CO₂ signals converge to regulate light-induced stomatal opening" by Hiyama et al., reports proteomic identification of a MAPKKK that is phosphorylated by phototropins in response to blue light. The authors go on to nicely demonstrate how CBC1 (and 2) regulate stomatal aperture.

Using a combination of in vivo (mass spectrometry in *phot1 phot2* mutant; 1b) and in vitro kinase assays (fig 4) the authors convincingly demonstrate the phototropins phosphorylate CBC1 on S43/S45. Additionally, in vitro assays show that HT1 phosphorylates CBC1 and CBC2 though the sites of phosphorylation remain elusive, which I agree is beyond the scope of this manuscript. While the phosphorylation data is solid I do have several comments:

- Details for proteomics methods are insufficient. What is the FDR used for database search, # missed cleavages allowed, fixed modifications, mass tolerance etc. Additionally, should provide a supplemental file of all peptides

identified, their MASCOT score, precursor charge and m/z, peptide identification score, which is standard for proteomic reports.

We rewrote details for proteomics analyses as below in Materials and Methods, and added the references.

The samples were suspended in 0.1 M Tris-HCl (pH 8.0) containing 8 M urea, protein phosphatase inhibitors, and protease inhibitor cocktails (SIGMA-Aldrich) according to the manufacture's protocol, and sonicated for 5 min. The suspensions were reduced with dithiothreitol, alkylated with iodoacetamide and digested with Lys-C, followed by tryptic digestion as described (Saito et al., 2006). The digested samples were desalted using StageTips with C18 Empore disk membranes (3M) (Rappsilber et al., 2003). Phosphopeptides were enriched by hydroxy acid-modified metal oxide chromatography (HAMMOc) using lactic acid-modified titania as described (Sugiyama et al., 2007; Kyono et al., 2008). In brief, the digested samples were diluted with 0.1% TFA, 80% acetonitrile containing 300 mg/mL lactic acid (solution A), and loaded to custom-made metal oxide chromatography tips preliminary equilibrated with solution A. After successive washing with solution A and 0.1% TFA, 80% acetonitrile, the peptides were eluted with 0.5% piperidine. The obtained fractions were desalted using SDB-XC-StageTips and concentrated in a vacuum evaporator for nanoLC-MS/MS analysis. NanoLC-MS/MS analyses were performed by using LTQ-Orbitrap (Thermo Fisher Scientific, Rockwell, Illinois, USA) with a previously described setup (Nakagami et al., 2010). Peptides and proteins were identified by means of automated database searching using Mascot version 2.3 (Matrix Science, London, UK) against the TAIR database (release 10) with a precursor mass tolerance of 3 ppm, a fragment ion mass tolerance of 0.8 Da, and strict trypsin specificity allowing for up to two missed cleavages. Cysteine carbamidomethylation was set as a fixed modification, and methionine oxidation and phosphorylation of serine, threonine, and tyrosine were allowed as variable modifications. Peptides were considered identified if the Mascot score was over the 95% confidence limit based on 'identity' score of each peptide and if at least three successive y or b ions with an additional two or more y, b ions were

observed. False-positive rate was evaluated by searching against a randomized decoy database created by the Mascot Perl script, and estimated at less than 1% for phosphopeptide identification in all LC-MS data. Phosphorylation sites were unambiguously determined when b- or y-ions were between which the phosphorylated residue exists were observed in the peak list of fragment ions. Based on peptide information (observed m/z of monoisotopic ion and retention time) obtained by database searching, the LC-MS peak area of each peptide in all samples was determined by integration of ion intensity in survey MS scan. The peak integration was performed by Gaussian approximation of an extracted ion chromatogram within 5 mDa of the observed m/z using Mass Navigator v1.2.

1. Saito, H., Oda, Y., Sato, T., Kuromitsu, J., and Ishihama, Y. (2006) Multiplexed two-dimensional liquid chromatography for MALDI and nanoelectrospray ionization mass spectrometry in proteomics. *J. Proteome Res.* 5, 1803-1807.
2. Rappsilber, J., Ishihama, Y., and Mann, M. (2003) Stop and go extraction tips for matrix-assisted laser desorption/ionization, nanoelectrospray, and LC/MS sample pretreatment in proteomics. *Anal Chem* 75, 663-670.
3. Sugiyama, N., Masuda, T., Shinoda, K., Nakamura, A., Tomita, M., and Ishihama, Y. (2007) Phosphopeptide enrichment by aliphatic hydroxy acid-modified metal oxide chromatography for nano-LC-MS/MS in proteomics applications. *Mol Cell Proteomics* 6, 1103-1109.
4. Kyono, Y., Sugiyama, N., Imami, K., Tomita, M., and Ishihama, Y. (2008) Successive and selective release of phosphorylated peptides captured by hydroxy acid-modified metal oxide chromatography. *J Proteome Res* 7, 4585-4593.
5. Nakagami, H., Sugiyama, N., Mochida, K., Daudi, A., Yoshida, Y., Toyoda, T., Tomita, M., Ishihama, Y., and Shirasu, K. (2010) Large-scale comparative phosphoproteomics identifies conserved phosphorylation sites in plants. *Plant Physiol* 153, 1161-1174.

We provided a supplemental excel file of all peptides identified, their MASCOT score, precursor charge and m/z, peptide identification score in fullPPIlist as supplementary Table 1.

Raw proteomics data needs to be deposited in a repository (MASSIVE, Proteome Exchange ect).

We upload the MS data in a repository of MassIVE. We mentioned it in Materials and Methods as “The MS data files are available from ProteomeXchange with the identifier PXD006586”.

No description of CBC1 or CBC2 antibodies.

We described it in Materials and Methods as: Generation of specific antibodies against CBC1 and CBC2. The polyclonal antibodies against the GST-tagged recombinant protein fragments of the respective CBC1 and CBC2 (CBC1: Asp²⁵ to Ile¹⁰³, CBC2: Gln²² to Ile⁷⁷) were raised in rabbits. The fragments were generated in *E. coli* as antigens, and were purified by glutathione-Sepharose beads with elution by reduced glutathione.

Panel 6F is not described in the figure legend.

Thank you. We noted this in the legend of Fig. 6f as “in vitro phosphorylation of CBC1 (D271N) and Ser-substituted CBC1 (D271N S43A S45A) by HT1”.

REVIEWERS' COMMENTS:

Reviewer #5 (Remarks to the Author):

The authors have thoroughly addressed by previous comments.